# UltraViCo: Breaking Extrapolation Limits in Video Diffusion Transformers

**Min Zhao**[1,2] [*]**, Hongzhou Zhu**[1,2] [*]**, Yingze Wang**[1] [*]**, Bokai Yan**[3]**, Jintao Zhang**[1,2]**, Guande He**[4]**, Ling Yang**[5]**, Chongxuan Li**[3]**, Jun Zhu**[1,2]

[1]Dept. of Comp. Sci. & Tech., BNRist Center, THU-Bosch ML Center, Tsinghua University.
[2]ShengShu. [3]Gaoling School of Artificial Intelligence, Renmin University of China.
[4]The University of Texas at Austin. [5] Princeton University.
`gracezhao1997@gmail.com, zhuhz22@mails.tsinghua.edu.cn`

## Abstract

Despite advances, video diffusion transformers still struggle to generalize beyond their training length, a challenge we term video length extrapolation. We identify two failure modes: model-specific *periodic content repetition* and a universal *quality degradation*. Prior works attempt to solve repetition via positional encodings, overlooking quality degradation and achieving only limited extrapolation. In this paper, we revisit this challenge from a more fundamental view—attention maps, which directly govern how context influences outputs. We identify that both failure modes arise from a unified cause: *attention dispersion*, where tokens beyond the training window dilute learned attention patterns. This leads to quality degradation and repetition emerges as a special case when this dispersion becomes structured into *periodic attention patterns*, induced by harmonic properties of positional encodings. Building on this insight, we propose *UltraViCo*, a training-free, plug-and-play method that suppresses attention for tokens beyond the training window via a constant decay factor. By jointly addressing both failure modes, we outperform a broad set of baselines largely across models and extrapolation ratios, pushing the extrapolation limit from $2\times$ to $4\times$. Remarkably, it improves Dynamic Degree and Imaging Quality by 233% and 40.5% over the previous best method at $4\times$ extrapolation. Furthermore, our method generalizes seamlessly to downstream tasks such as controllable video synthesis and editing. Project page is available at https://thu-ml.github.io/UltraViCo.github.io/.

## 1 Introduction

Building upon the expressive power of diffusion transformers (DiTs) (Bao et al., 2023; Peebles & Xie, 2023), recent advances in text-to-video (T2V) generation Bao et al. (2024); Zheng et al. (2024b); Brooks et al. (2024); Wan et al. (2025); Kong et al. (2024); Hong et al. (2022) have enabled models to synthesize high-fidelity videos. However, these models are typically trained on a fixed maximum sequence length (e.g., 5 seconds Wan et al. (2025); Kong et al. (2024); Hong et al. (2022)) and struggle to generate videos beyond their training length, a task we term *video length extrapolation*, which is critical for practical applications.

To investigate the core challenges of this task, we conduct experiments on a range of models and identify two failure modes: (i) a model-specific *periodic content repetition*, where short clips loop indefinitely in certain models; and (ii) a universal *quality degradation*, manifested as blurred spatial details and frozen temporal dynamics across all models. Both failures become increasingly severe as the extrapolation length grows. Prior work, such as RIFLEx (Zhao et al., 2025), tackles repetition from the perspective of positional encodings, while overlooking quality degradation and therefore achieving limited extrapolation. We contend, however, that positional encodings play only an *indirect* role by perturbing queries and keys to influence attention. In contrast, attention itself—*directly* aggregating contextual information to generate outputs—offers a more fundamental view.

---

[*]Equal contribution.

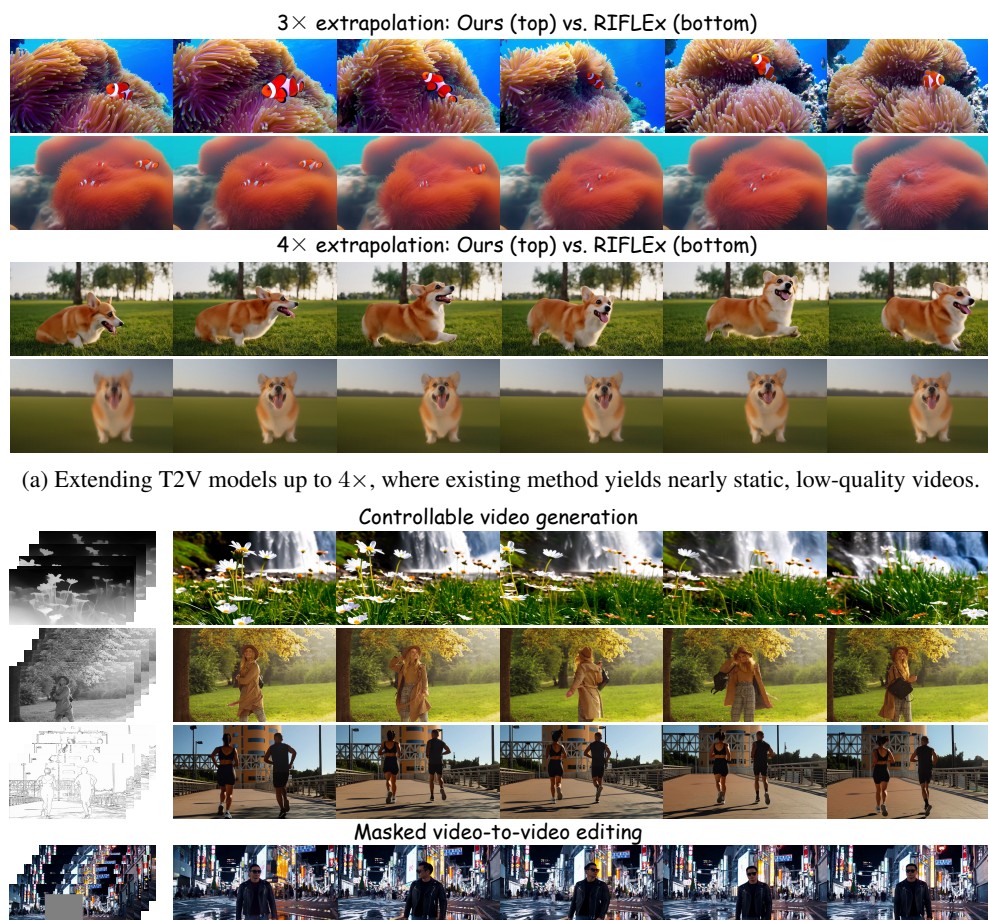

3× extrapolation: Ours (top) vs. RIFLEx (bottom)

4× extrapolation: Ours (top) vs. RIFLEx (bottom)

(a) Extending T2V models up to 4×, where existing method yields nearly static, low-quality videos.

Controllable video generation

Masked video-to-video editing

(b) Generalization to downstream tasks at 3×. See more tasks in Appendix C.4.

Figure 1: **Visual results.** UltraViCo achieves significant extrapolation improvement on (a) T2V models and (b) downstream tasks. *See prompts and videos in supplementary materials.*

Therefore, we revisit extrapolation failures through the lens of attention maps. Our systematic analysis of attention maps shows that both failure modes arise from a unified mechanism: *attention dispersion*. This occurs when new tokens beyond the training length dilute the learned attention patterns. This leads to quality degradation and repetition arises as a special case when dispersion becomes organized into *periodic attention patterns*. Specifically, this happens when positional encoding frequencies form *harmonics*, enabling the largest-amplitude frequency and its harmonics to accumulate amplitude and contribute substantially to the overall amplitude.

Building on this unified view, we propose ***Ul**tra*-extrapolated ***Vi**deo via Attention **Co**ncentration (***UltraViCo***), a plug-and-play method that suppresses attention for tokens beyond the training window with a constant decay factor. This adjustment reallocates attention to reliable in-window context while naturally breaking periodic patterns, thus simultaneously addressing both failure modes. Notably, standard attention implementations encounter out-of-memory errors when modifying logits for long video sequences. We therefore develop a memory-efficient CUDA kernel that enables scalable applications on large video models.

To validate our approach, we conduct comprehensive evaluations on various T2V models (Kong et al., 2024; Yang et al., 2024; Wan et al., 2025) and extrapolation ratios, against a large family of baselines (Chen et al., 2023b; bloc97, 2023; Zhuo et al., 2024; Peng et al., 2023; Zhao et al., 2025). Experiments demonstrate that our method consistently surpasses all baselines in all settings by simultaneously addressing both failure modes. Notably, while prior methods collapse beyond 3× extrapolation and yield static videos, ours maintains fluid motion, effectively extending the practical limit from 2× to 4×. Remarkably, it improves Dynamic Degree and Imaging Quality by 233% and

40.5% over the previous best method at $4\times$ extrapolation. Beyond this, our method also generalizes seamlessly to downstream tasks such as various controllable video synthesis and editing.

Figure 2: **Failure modes of video length extrapolation.** Some models exhibit *periodic content repetition*, while *quality degradation* occurs universally. Both failure modes intensify with longer extrapolations. "extra." denotes extrapolation. See Appendix C.1 for additional models.

## 2 PRELIMINARY

**Attention mechanism with rotary position embedding.** Modern video diffusion models are largely built on DiTs whose core is the attention mechanism (Vaswani et al., 2017; Li et al., 2025a). The input video is patched into $L$ tokens, each projected into queries, keys, and values. To encode the position information, DiTs mainly adopt Rotary Position Embedding (RoPE) (Su et al., 2024), which injects position into queries and keys through complex rotations. Concretely, for each query or key vector $\boldsymbol{x} \in \mathbb{R}^D$ at position $t$, RoPE maps it to $\mathbb{R}^D$ as

$$\boldsymbol{f}^{\text{RoPE}}(\boldsymbol{x}, t)_i = R_i(t) \begin{bmatrix} x_{2i} \\ x_{2i+1} \end{bmatrix}, \; R_i(t) = \begin{bmatrix} \cos(\phi_i t) & -\sin(\phi_i t) \\ \sin(\phi_i t) & \cos(\phi_i t) \end{bmatrix}, \; i \in \{0, \ldots, D/2 - 1\}. \quad (1)$$

Here, each frequency $\phi_i$ depends exponentially on $i$ and is used to encode the $(2i, 2i+1)$ components of $\boldsymbol{x}$. After RoPE, the queries and keys form matrices $\boldsymbol{Q} \in \mathbb{R}^{L \times D}$ and $\boldsymbol{K} \in \mathbb{R}^{L \times D}$. Their interaction yields the attention logits $\boldsymbol{S} \in \mathbb{R}^{L \times L}$, which are normalized by the softmax function to obtain the attention scores $\boldsymbol{P} \in \mathbb{R}^{L \times L}$. These scores are then applied to the value matrix $\boldsymbol{V} \in \mathbb{R}^{L \times D'}$ to produce the output $\boldsymbol{O} \in \mathbb{R}^{L \times D'}$:

$$\boldsymbol{S} = \boldsymbol{Q}\boldsymbol{K}^\top, \quad \boldsymbol{P} = \text{softmax}(\frac{\boldsymbol{S}}{\sqrt{D}}), \quad \boldsymbol{O} = \boldsymbol{P}\boldsymbol{V}. \quad (2)$$

For videos with temporal and spatial axes, Multimodal RoPE (M-RoPE) (Wang et al., 2024a) partitions the dimension $D = d_\mathcal{T} + d_\mathcal{H} + d_\mathcal{W}$ and encodes each subspace separately. Since we focus on temporal extrapolation, we consider only the temporal axis and denote $d_\mathcal{T}$ as $d$ for simplicity (see details in Appendix B.2).

**Problem setting: video length extrapolation.** Despite advances, DiT-based video generation models struggle to produce videos longer than their training duration. This task, known as *video length extrapolation* (Zhao et al., 2025), aims to adapt a pre-trained model to generate high-quality videos of a sequence length $L'$ that exceeds its training length $L$, with the extrapolation ratio defined as $s = L'/L > 1$. Notably, video length extrapolation targets the model's intrinsic ability to generate longer sequences in a single forward generation, which is orthogonal to prior methods (Qiu et al., 2023; Wang et al., 2023; Kim et al., 2024; Wang et al., 2024c; Lu et al., 2024) that rely on inference-time modifications. See Appendix A for more related work.

## 3 METHOD

### 3.1 FAILURE MODES OF VIDEO LENGTH EXTRAPOLATION

In this section, we investigate the core challenges of video length extrapolation on a range of SOTA video diffusion transformers, including Wan (Wan et al., 2025), HunyuanVideo (Kong et al., 2024), and CogVideoX (Yang et al., 2024).

Qualitative results in Fig.2a and Fig.2b reveal two distinct failure modes. The first is a *periodic content repetition*, which occurs in certain models such as HunyuanVideo and CogVideoX. The second is a universal *quality degradation*, characterized by compromised spatial fidelity and temporal dynamics across all models. To further investigate their trends across extrapolation lengths, we perform a quantitative analysis on 10 prompts using metrics including Imaging Quality (Huang et al., 2024), Dynamic Degree (Huang et al., 2024), and Repetition Count. Fig. 2c confirms that both failures become more severe as the extrapolation factor increases.

These findings raise three critical questions: First, *why does periodic content repetition only manifest in specific models?* Second, *what is the underlying cause of the universal quality degradation?* Most importantly, *is there a unified cause behind these two seemingly independent failure modes?*

Existing work such as RIFLEx addresses only content repetition, neglecting quality degradation, which limits both model generalization and extrapolation capacity. While RIFLEx attributes repetition to positional encoding periodicity, we argue that positional encodings play only an indirect role by modulating queries and keys. Instead, as Eq. (2) shows, the attention map itself is fundamental, since it directly determines how context is aggregated. This motivates us to revisit extrapolation failures through attention analysis.

### 3.2 ATTENTION ANALYSIS OF THE CAUSE

In this section, we first focus on the specific issue of periodic content repetition (Sec. 3.2.1). Through an in-depth attention analysis of its underlying mechanism, we find, surprisingly, that the solution designed to resolve repetition also improves video quality. This key finding then allows us to understand the cause of the more universal problem of quality degradation (Sec. 3.2.2), and ultimately reveals the intrinsic connection between the two failure modes.

#### 3.2.1 THE CAUSE OF CONTENT REPETITION: PERIODIC ATTENTION PATTERNS

**Periodic attention induces output repetition.** We analyze the cause of content repetition by inspecting the attention map $\boldsymbol{P} \in \mathbb{R}^{L' \times L'}$ during $4\times$ extrapolation, where $L'$ is the extrapolated sequence length (i.e., video features flattened into a 1D sequence). The entry at row $i$, column $j$ of $\boldsymbol{P}$, denoted $P_{ij}$, is the attention score from query $i$ to key $j$. As shown in Fig. 3a, the attention map of HunyuanVideo reveals two properties that jointly induce periodic outputs.

First, the map exhibits a distinct *row-wise periodicity*. Specifically, for any query at position $i$, its attention scores to key positions $j$ and $j+T$ are nearly identical: $\boldsymbol{P}_{i,j} \approx \boldsymbol{P}_{i,j+T}$, where $T$ corresponds to the observed repetition period in Sec. 3.1. As indicated in Fig. 3a, the blue and purple circles highlight nearly equal scores. Second, the map shows *relative positional invariance*: query–key pairs with the same relative displacement $p$ yield approximately equal scores, $\boldsymbol{P}_{i,j} \approx \boldsymbol{P}_{i+p,j+p}$. This RoPE-induced property appears as uniform values along diagonals and subdiagonals; for example, when $p = T$, the scores marked by the blue and green circles are nearly identical.

Combining these properties, we can derive that entire query rows also repeat periodically: $\boldsymbol{P}_{i+T,j} \approx \boldsymbol{P}_{i,j}$, as shown by the green and purple circles. Thus, rows $i$ and $i + T$ retrieve nearly the same weighted information from the value $\boldsymbol{V}$, leading to periodic outputs (see Appendix B.1 for details):

$$\boldsymbol{O}_{i+T} = \sum_{j=0}^{L'-1} \boldsymbol{P}_{i+T,j} \boldsymbol{V}_j \approx \sum_{j=0}^{L'-1} \boldsymbol{P}_{i,j} \boldsymbol{V}_j = \boldsymbol{O}_i. \tag{3}$$

This periodicity is directly reflected in repeated content in pixel space. Larger extrapolation ratios traverse more periods, thus increasing repetition counts, which is consistent with our observations in

| Model | Attention maps | Statistical row-wise attention analysis |
|---|---|---|

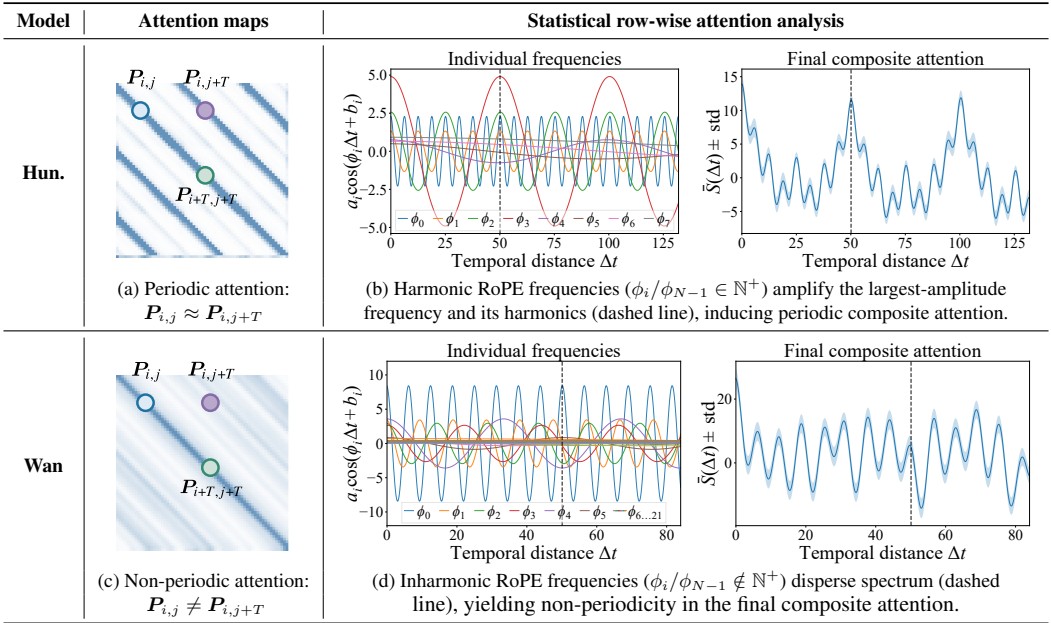

(a) Periodic attention: $\boldsymbol{P}_{i,j} \approx \boldsymbol{P}_{i,j+T}$

(b) Harmonic RoPE frequencies ($\phi_i/\phi_{N-1} \in \mathbb{N}^+$) amplify the largest-amplitude frequency and its harmonics (dashed line), inducing periodic composite attention.

(c) Non-periodic attention: $\boldsymbol{P}_{i,j} \neq \boldsymbol{P}_{i,j+T}$

(d) Inharmonic RoPE frequencies ($\phi_i/\phi_{N-1} \notin \mathbb{N}^+$) disperse spectrum (dashed line), yielding non-periodicity in the final composite attention.

Figure 3: **Periodic attention patterns as cause of content repetition.** Left: unlike Wan, HunyuanVideo exhibits row-wise periodic attention during $4\times$ extrapolation, causing repeated outputs. Right: statistical row-wise attention can be expressed as a linear combination of trigonometric functions of RoPE frequencies, whose properties govern this periodicity. Hun. denotes HunyuanVideo.

Sec. 3.1. By contrast, the attention map of Wan (Fig. 3c) does not display such row-wise periodicity, and accordingly its outputs remain free of repetition.

**Origin of periodic attention patterns.** Next, we show that such model-specific row-wise periodicity originates from the RoPE frequencies. To reveal the core row-wise attention structure from noise, we construct a statistical row attention pattern $\bar{\boldsymbol{S}}(\Delta t)$, which captures the relation between a query and keys at the same spatial location but $\Delta t$ latent frames apart. This is achieved by taking the expectation of the pre-softmax attention logits across all layers, heads, and query positions. As derived in Appendix B.3 (based on Eq. (2)), this quantity admits the following trigonometric decomposition:

$$\bar{\boldsymbol{S}}(\Delta t) = \sum_{i=0}^{d/2-1} a_i \cos(\phi_i \Delta t + b_i) + C, \tag{4}$$

where $\{\phi_i\}_{i=0}^{d/2-1}$ are the RoPE frequencies defined in Sec. 2, and $\{a_i\}_{i=0}^{d/2-1}$, $\{b_i\}_{i=0}^{d/2-1}$, $C$ are constants determined by the statistics of queries and keys from models, with $b_i$ typically close to zero. Visualizations of these frequency components for HunyuanVideo and Wan highlight a crucial difference (Fig. 3b,d, left). The periodicity of such a superposition is decided by the frequency relationships, as formalized in Proposition 1.

**Proposition 1** (Period and Amplitude of Harmonics). *For a function $f(\Delta t) = \sum_{i=0}^{N-1} a_i \cos(\phi_i \Delta t)$, where $a_i > 0, \phi_i > 0$ and $\min_i \phi_i = \phi_{N-1}$, if and only if $\forall i, \phi_i/\phi_{N-1} \in \mathbb{N}^+$ (i.e., they form a set of **harmonics**), $f(\Delta t)$ is periodic with period $T_{N-1} = \frac{2\pi}{\phi_{N-1}}$. In this case, $\max_{\Delta t} f(\Delta t) = \sum_{i=0}^{N-1} a_i$, whenever $\Delta t = mT_{N-1}$, $m \in \mathbb{Z}$ (i.e., whenever $\Delta t$ is at **harmonic alignment positions**).*

We find that HunyuanVideo's frequencies satisfy this *harmonic* condition in Proposition 1, allowing amplitude accumulation of the largest-amplitude frequency $\phi_3$ and its harmonics ($i < 3$) at *harmonic alignment positions* $mT$ (dashed line in Fig. 3b), where $m \in \mathbb{Z}$. This yields a dominant component that contributes 79.6% of the total amplitude, producing a strongly periodic composite attention pattern (Fig. 3b, right). A similar harmonic alignment is also observed in CogVideoX (Appendix B.6). In contrast, Wan's frequencies are not harmonically aligned, resulting in a dispersed spectrum where no frequency dominates (largest 31.6%), and thus no clear periodicity emerges (Fig. 3d). Notably,

while the strict periodicity of HunyuanVideo is determined by the lowest frequency, its small amplitude and long period make it negligible; the observed periodicity $T$ is effectively governed by the dominant frequency (see Appendix B.6).

In summary, our analysis establishes the causal chain: *RoPE-induced frequency harmonics lead to periodic attention patterns, which in turn produce periodic output features and ultimately manifest as content repetition*. To validate this, we mask tokens at harmonic alignment positions $mT$. Breaking these constructive interference points disrupts periodic attention and, as shown in Fig. 4a, effectively mitigates repetition.

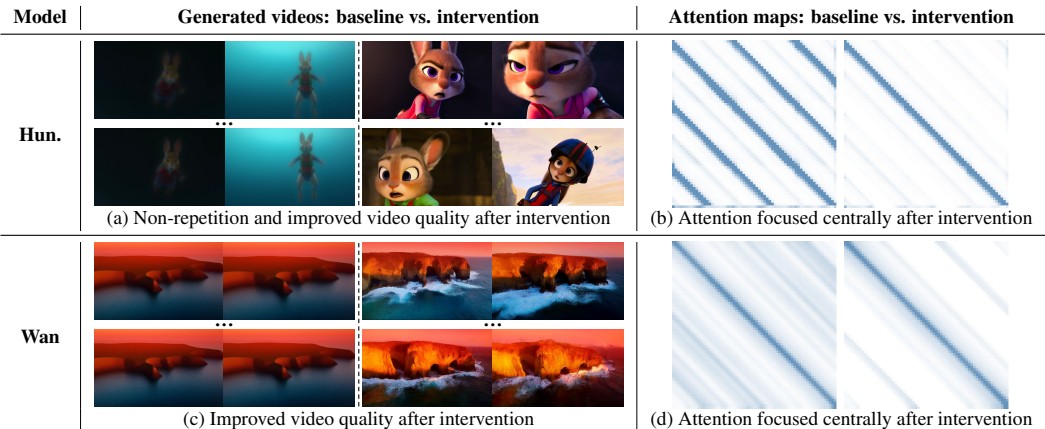

| Model | Generated videos: baseline vs. intervention | Attention maps: baseline vs. intervention |
|---|---|---|
| Hun. | (a) Non-repetition and improved video quality after intervention | (b) Attention focused centrally after intervention |
| Wan | (c) Improved video quality after intervention | (d) Attention focused centrally after intervention |

Figure 4: **Fixing repetition reveals attention dispersion as the fundamental cause.** Left: our intervention, initially targeting repetition, surprisingly enhances video quality in both models. Right: the shared mechanism is revealed, where the intervention refocuses diffuse baseline attention toward the central training window. This suggests attention dispersion as the unified cause.

### 3.2.2 THE CAUSE OF QUALITY DEGRADATION: ATTENTION DISPERSION

Surprisingly, we find the above repetition-resolving intervention also improves video quality across both models (Fig. 4a, c). This finding suggests a more profound hypothesis: content repetition and quality degradation may arise from a shared, fundamental underlying mechanism.

A comparison of attention maps shows our intervention consistently concentrates the initially diffuse attention (Fig. 4b, d). This occurs because masking the harmonic peaks forces a softmax renormalization, which sharpens the attention distribution by proportionally increasing the remaining scores. To further identify where this sharpened focus is most beneficial, we systematically masked different attention regions and found that concentrating attention within the original central training window yielded the strongest improvements (see details in Appendix B.7). This leads us to hypothesize that *attention dispersion* is the underlying issue. New tokens during extrapolation dilute the learned attention patterns within the original training window. This dispersion has two detrimental effects. Spatially, the model needs to consider far-away extrapolated frames, which makes it difficult to focus on fine details and results in visual blurriness. Temporally, taking these distant frames into account mixes local motion with unrelated movements, causing the video to appear static and unnatural. These effects are consistent with the quality degradation observed in Sec. 3.1.

To validate this hypothesis, we conduct a controlled experiment where we progressively mask attention scores for tokens outside the training window, thereby forcing the attention to concentrate centrally. The results, presented in Fig. 5, demonstrate a clear positive correlation: more concentrated attention (i.e., by increasing the proportion of masked out-of-window scores) consistently improves both the visual quality and motion dynamics of the generated video. This provides strong evidence that attention dispersion is the cause of quality degradation. Consequently, as the extrapolation ratio increases, attention becomes more dispersed, leading to worse quality, consistent with the observations in Sec. 3.1.

**A unified view: periodic attention as a case of attention dispersion.** Building upon the above analysis, we can unify both failure modes under a single perspective: attention dispersion is the fundamental cause of extrapolation failure, with periodic attention patterns representing a special

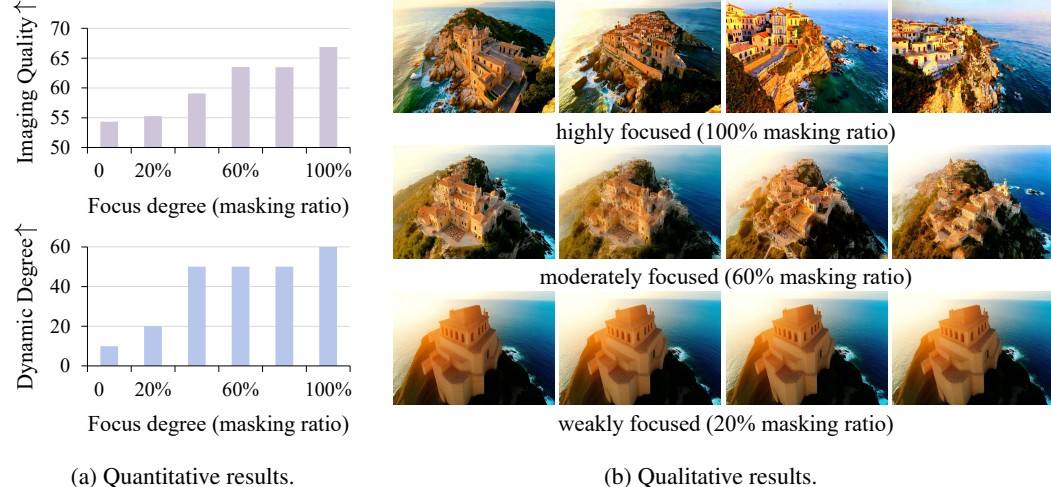

Figure 5: **Validation of attention dispersion as the cause of quality degradation.** Both (a) quantitative and (b) qualitative results show that video quality improves monotonically as the degree of attention central focusing (i.e., the masking ratio of out-of-window scores) increases.

case. Specifically, when a RoPE frequency contributes substantially to the overall amplitude (e.g., due to harmonic alignment), it induces a strongly periodic attention pattern; otherwise, the model exhibits generic, non-periodic dispersion.

### 3.3 ULTRAVICO

Building on the above unified view, we propose **Ultra**-extrapolated **V**ideo via Attention **Co**ncentration (**UltraViCo**), a simple yet effective method that suppresses attention for tokens beyond the training window via a decay factor, thereby restoring the model's focusing ability. To achieve this, we introduce a position-dependent decay factor $\lambda_{ij}$ applied to the original attention logits $S_{ij}$, yielding the corrected attention $S'_{ij}$:

$$S'_{ij} = \lambda_{ij} \cdot S_{ij}, \quad \text{where} \quad \lambda_{ij} = \begin{cases} 1, & \text{if } |i - j| \leq L/2 \text{ or } S_{ij} < 0, \\ \alpha, & \text{otherwise}, \end{cases} \quad (5)$$

where $\alpha < 1$ is a constant decay hyperparameter and $L$ is the training length. Here, $\lambda_{ij}$ is set to be 1 for all pairs within the training window, preserving the model's core learned dynamics. For out-of-window tokens, only positive logits ($S_{ij} \geq 0$) are down-scaled because multiplying negative logits $S_{ij} < 0$ by $\alpha < 1$ can undesirably increase its value, while multiplying $\alpha > 1$ or 1 for negative logits has a negligible effect. We also experimented with various decay strategies, such as linear decay, but found the constant form is sufficient, indicating that the key is distinguishing in-window from out-of-window tokens rather than the decay shape itself (see Sec. 4.2 for details).

However, in models showing periodic repetition (Sec. 3.2.1), harmonic alignment positions $mT$ attract disproportionately high attention. Applying a uniform small decay $\alpha$ would overly suppress all out-of-window context, harming temporal consistency. To address this, we apply a stronger decay $\beta < \alpha$ specifically to these risky positions $mT$, while keeping $\alpha$ for other out-of-window tokens:

$$\lambda_{ij} = \begin{cases} 1, & \text{if } |i - j| \leq L/2 \text{ or } S_{ij} < 0, \\ \beta, & \text{else if } (i, j) \in \mathcal{P}_{\text{risk}}, \\ \alpha, & \text{otherwise}, \end{cases} \quad (6)$$

where $\mathcal{P}_{\text{risk}} = \left\{ (i, j) \mid mT - \gamma \leq i - j \leq mT + \gamma, \ m \in \mathbb{Z}, \gamma \in \mathbb{N}^+ \right\}$ denotes the set of positions within $\gamma$ frames around the harmonic alignment positions $mT$ and $\beta < \alpha < 1$. This targeted adjustment reallocates attention to reliable in-window context while eliminating spurious periodic patterns, allowing UltraViCo to mitigate both failure modes simultaneously.

**Efficient CUDA implementation.** UltraViCo requires modifying attention logits, but standard PyTorch attention is infeasible for long sequences. At a $3\times$ extrapolation ($\sim$200K tokens for Hunyuan-

Table 1: **Quantitative illustrative results on VBench for HunyuanVideo and Wan.** For Wan, which does not exhibit content repetition, we omit the NoRepeat Score. Additional results for more extrapolation ratios and models are provided in Appendix C.3. Consist., Dyn., Qual., Over. and NoRe. denote Consistency, Dynamics, Quality, Overall and NoRepeat Score respectively. Normal. indicates the training length for reference.

| Method | Wan2.1-1.3B | | | | | HunyuanVideo | | | | | |
|---|---|---|---|---|---|---|---|---|---|---|---|
| | Consist.↑ | Dyn.↑ | Qual.↑ | Over.↑ | User↓ | Consist.↑ | NoRe.↑ | Dyn.↑ | Qual.↑ | Over.↑ | User↓ |
| Normal. | 0.9554 | 51 | 70.34 | 24.25 | – | 0.9786 | – | 71 | 69.31 | 26.81 | – |
| 3× extrapolation | | | | | | | | | | | |
| PE | 0.9419 | 6 | 56.28 | 18.53 | 3.82 | 0.9795 | 53.17 | 16 | 51.85 | 21.62 | 3.96 |
| PI | 0.9667 | 7 | 52.16 | 17.48 | 4.69 | 0.9787 | 90.23 | 1 | 46.30 | 21.29 | 4.91 |
| NTK | 0.9437 | 3 | 57.73 | 18.50 | 4.40 | 0.9802 | 84.80 | 24 | 53.11 | 22.14 | 3.74 |
| YaRN | **0.9676** | 5 | 53.46 | 17.53 | 4.71 | 0.9790 | 88.74 | 0 | 47.05 | 21.42 | 5.05 |
| TASR | 0.9434 | 6 | 57.41 | 18.48 | 4.47 | 0.9807 | 80.74 | 22 | 51.95 | 22.02 | 4.65 |
| RIFLEx | 0.9431 | 5 | 53.79 | 17.54 | 4.90 | **0.9823** | 73.97 | 17 | 50.57 | 21.22 | 4.67 |
| **Ours** | 0.944 | **46** | **62.43** | **23.21** | **1.01** | 0.9465 | **100.0** | **62** | **65.00** | **26.45** | **1.02** |
| 4× extrapolation | | | | | | | | | | | |
| PE | 0.9415 | 11 | 55.25 | 16.65 | 3.75 | 0.9891 | 31.41 | 14 | 47.12 | 17.61 | 3.70 |
| PI | 0.9711 | 12 | 50.44 | 16.34 | 4.87 | 0.9885 | 70.93 | 0 | 42.19 | 17.83 | 4.82 |
| NTK | 0.9477 | 11 | 55.37 | 16.09 | 4.24 | **0.9915** | 72.39 | 10 | 50.01 | 18.92 | 4.23 |
| YaRN | **0.9729** | 7 | 51.16 | 16.69 | 4.57 | 0.9877 | 62.87 | 1 | 41.37 | 18.53 | 5.03 |
| TASR | 0.9495 | 9 | 55.18 | 16.16 | 4.72 | 0.9911 | 51.28 | 14 | 46.81 | 18.47 | 4.51 |
| RIFLEx | 0.9453 | 10 | 51.05 | 15.83 | 4.84 | 0.9906 | 52.84 | 11 | 41.02 | 16.47 | 4.69 |
| **Ours** | 0.9484 | **47** | **59.36** | **21.61** | **1.01** | 0.9468 | **99.87** | **42** | **66.54** | **24.52** | **1.02** |

Video), for instance, materializing a 200K × 200K attention mask consumes over 80GB of memory in `bf16`, causing an immediate out-of-memory error. To address this, we integrate UltraViCo into Triton-based FlashAttention (Dao et al., 2022) and SageAttention (Zhang et al., 2024b), where the online-softmax formulation avoids explicit mask construction. This yields scalable, memory-efficient computation, enabling UltraViCo on large video models.

## 4 EXPERIMENTS

### 4.1 SETUP

**Evaluation.** We evaluate methods on three video diffusion models, including HunyuanVideo, Wan2.1-1.3B and CogVideoX-5B. Following RIFLEx, we use 100 prompts sampled from VBench (Huang et al., 2024). For quantitative evaluation, following RIFLEx, we adopt Imaging Quality (Quality), Dynamic Degree (Dynamics), and Overall Consistency (Overall) from VBench, along with the NoRepeat Score for models prone to content repetition. Notably, our NoRepeat Score is a variant of that in RIFLEx, tailored for multiple-repetition (see Appendix C.2 for details). Finally, we conduct a user study with 10 participants on 10 prompts, where users rank (User) the overall quality of videos across all methods. More details are provided in Appendix C.2.

**Implementation Details.** The decay factor $\alpha$ is set to 0.9 for Wan and HunyuanVideo at 3× and 4× extrapolation. For HunyuanVideo, we set $\gamma = 4$ for all ratios, and $\beta = 0.6$ at 3× and 0.8 at 4×. Our baseline configurations follow RIFLEx. Further details are provided in Appendix C.2.

### 4.2 RESULTS

**Performance comparison.** We compare a wide range of length extrapolation baselines on three SOTA models (Kong et al., 2024; Yang et al., 2024; Wan et al., 2025) across various extrapolation ratios, including PE (Zhao et al., 2025), PI (Chen et al., 2023b), NTK (bloc97, 2023), TASR (Zhuo et al., 2024), YaRN (Peng et al., 2023), and RIFLEx. Tab. 1 reports 3× and 4× results on Hunyuan-Video and Wan, while Fig. 6 shows qualitative samples on HunyuanVideo. Results for additional ratios and models are provided in the Appendix C.3.

As shown in Tab. 1, our method consistently outperforms all baselines across models and extrapolation ratios, simultaneously improving video quality and eliminating content repetition. Specifi-

cally, PE suffers from severe repetition, reflected in low NoRepeat Scores. In contrast, our method achieves substantially higher scores, effectively removing repetition. Beyond repetition, unlike RI-FLEx which targets only this issue, our method delivers broader gains in both visual quality and motion quality. For instance, it improves Dynamic Degree and Imaging Quality on HunyuanVideo by 233% and 40.5% over the previous best method at $4\times$ extrapolation, respectively. Notably, on Wan beyond $3\times$ extrapolation, while prior methods collapse and yield static videos (Dynamic Degree $\leq 12$), our method restores fluid motion. By addressing both core failure modes, our method extends the extrapolation limit from $2\times$ to $4\times$. These improvements are further corroborated by user rankings (Tab. 1) and qualitative visualizations (Fig. 6), which consistently confirm the superior quality of our generated videos over baselines.

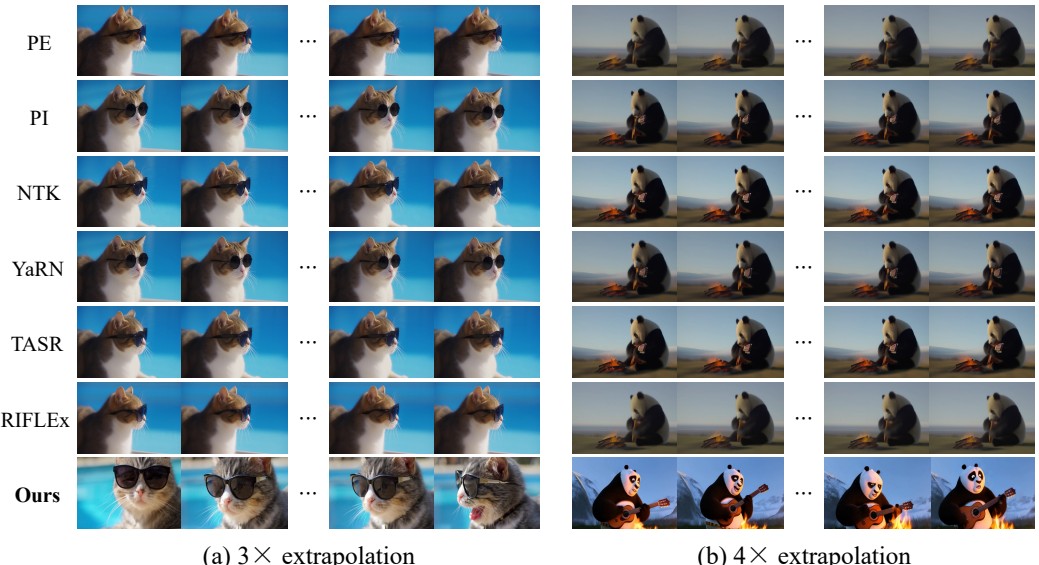

(a) $3\times$ extrapolation  (b) $4\times$ extrapolation

Figure 6: **Qualitative results on HunyuanVideo**. The baselines produce nearly static videos with poor visual quality, whereas our method achieves significantly better quality by addressing extrapolation failure modes. Additional qualitative results for other models are in Appendix C.4.

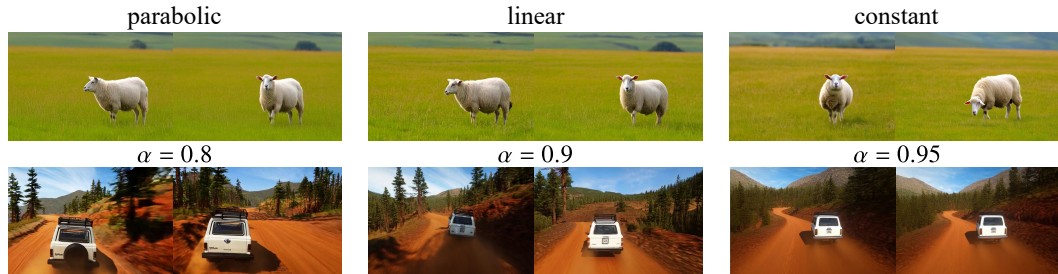

Figure 7: **Ablation studies.** Top row: different decay strategies have minor impact, suggesting simple constant decay suffices. Bottom row: small $\alpha$ harms consistency while large $\alpha$ offers limited gains. An intermediate value ($\alpha = 0.9$) enhances quality while preserving consistency.

**Ablation studies.** We ablate the decay strategy and the decay factor $\alpha$ on Wan at $3\times$ extrapolation. As shown in Fig. 7 (top), different decay strategies yield minor differences, indicating that simple constant decay suffices. As shown in Fig. 7 (bottom), strong decay harms consistency (i.e., the spare tire of the car disappears) while weak decay offers limited gains. An intermediate value ($\alpha = 0.9$) enhances quality while preserving consistency. Further details are provided in Appendix C.2. A sensitivity analysis for $\alpha$ and $\beta$ (Fig. 8) shows a stable trend: $\alpha \geq 0.9$ and $\beta \geq 0.6$ improve visual quality and motion dynamics while keeping temporal consistency near baseline. We adopt $\alpha = 0.9$ and $\beta = 0.6$ as robust defaults, with small adjustments possible (e.g., $\beta = 0.8$ for stronger consistency, $\alpha = 0.85$ for better quality). Although larger $\alpha$ and $\beta$ may introduce a mild reduction

in consistency, values above $0.94$ remain visually stable, aligning with common long-video settings (e.g., Wan's training-horizon consistency $\approx 0.95$). See more metrics of $\alpha, \beta$ in Tab. 4, 5, 6, and Fig. 17.

**Connection with other long-video generation methods.** UltraViCo aims to extend the effective training window of video diffusion transformers and is therefore orthogonal to existing long-video generation techniques such as FreeNoise (Qiu et al., 2023), FIFO-Diffusion (Kim et al., 2024), and sliding-window. As demonstrated in Table 2, enlarging the context window via UltraViCo consistently improves the long-term temporal consistency of these methods, without negatively affecting other performance. In Table 2, all methods follow the same evaluation setup ($6\times$ extrapolation for 30-second videos on Wan), where UltraViCo extends the base model's training window by $3\times$.

**Generalization to downstream tasks.** Our method enhances the model's inherent ability to handle longer sequences, making it naturally applicable to downstream tasks. As shown in Fig. 1, based on VACE (Jiang et al., 2025b), UltraViCo enables $3\times$ extrapolation in controllable generation and video editing. See Appendix C.4 for additional results.

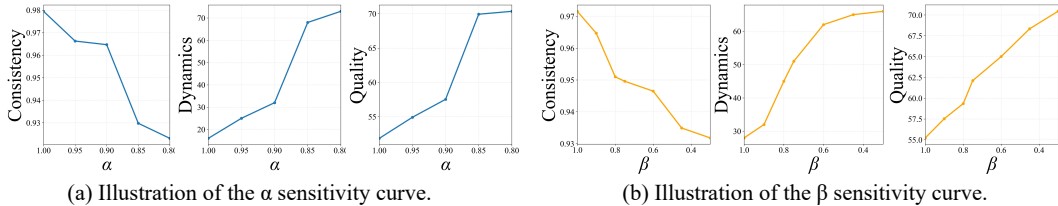

(a) Illustration of the α sensitivity curve.     (b) Illustration of the β sensitivity curve.

Figure 8: **Illustration of the hyperparameter sensitivity curve.** (a) When $\alpha \geq 0.9$, motion dynamics improve while consistency stays stable; below 0.9, consistency drops sharply. (b) When $\beta \geq 0.6$, dynamics remain high with comparable consistency; below 0.6, consistency degrades significantly.

Table 2: **Application of UltraViCo on existing long-video methods.**

| Method | Consistency↑ | Dynamics↑ | Quality↑ | Overall↑ |
|---|---|---|---|---|
| Sliding Window | 0.8478 | 56 | 62.94 | 23.57 |
| + UltraViCo | **0.9183** | 54 | 62.85 | 23.95 |
| FreeNoise | 0.9243 | 38 | 63.09 | 23.75 |
| + UltraViCo | **0.9431** | 41 | 62.12 | 23.92 |
| FIFO-Diffusion | 0.9131 | 53 | 61.31 | 23.81 |
| + UltraViCo | **0.9319** | 51 | 63.09 | 24.24 |

## 5    CONCLUSION

In this paper, we identify attention dispersion as the unified cause behind video length extrapolation failures. Based on this insight, we propose a training-free method that suppresses attention scores for tokens beyond training length. Experiments show that it significantly improves video quality, extending the practical extrapolation limit from $2\times$ to $4\times$.

## ETHICS STATEMENT

This paper advances the field of video generation, while emphasizing the importance of responsible use to avoid potential negative societal impacts, such as the creation of misleading or harmful content.

ACKNOWLEDGEMENTS

This work was supported by Fundamental and Interdisciplinary Disciplines Breakthrough Plan of the Ministry of Education of China (No. JYB2025XDXM101); National Natural Science Foundation of China (Nos. 62522609, 92470118); the Beijing Natural Science Foundation (No. L247030); and the fund for building world-class universities (disciplines) of Renmin University of China.

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

## USE OF LARGE LANGUAGE MODELS

We used a large language model solely to assist in polishing English writing and improving clarity. All research ideas, experiments, results, and interpretations are entirely our own.

## A  RELATED WORK

**Text-to-video Diffusion Transformers.**  Beyond autoregressive approaches Yan et al. (2025a;b), the recent advances in text-to-video generation have been primarily driven by diffusion models (Ho et al., 2020; Song et al., 2020; Ho et al., 2022; He et al., 2022; Zhao et al., 2022; 2023; Blattmann et al., 2023; Xing et al., 2023; Chen et al., 2023a; Zhao et al., 2024; Polyak et al., 2024; Zhou et al., 2024; Team, 2024a; Chen et al., 2024b; Ma et al., 2024a;b; 2025). With the development of diffusion transformers (DiTs) (Bao et al., 2023; Peebles & Xie, 2023), DiT-based text-to-video diffusion models have achieved remarkable performance, such as Sora (Brooks et al., 2024), Vidu (Bao et al., 2024), CogVideoX (Yang et al., 2024) and Open-Sora (Zheng et al., 2024a). Although achieving high quality, leading models are trained only on a fixed maximum sequence length, limiting long-term capacity. During video length extrapolation, they suffer from repetition or quality degradation, underscoring the need for length extrapolation.

**Length Extrapolation in Transformers.**  The goal of length extrapolation is to enable transformers to generate sequences longer than those seen during training in a single forward (Press et al., 2021). This is typically achieved by modifying positional encodings. For example, position interpolation (PI) (Chen et al., 2023b) improves performance by interpolating the frequencies in RoPE so that they remain within the training range even under extrapolation. NTK (bloc97, 2023),

YaRN (Peng et al., 2023), and Time-aware Scaled RoPE (TASR) (Zhuo et al., 2024) combine interpolation with direct extrapolation, incorporating adjustments along the token dimension, denoising timesteps, and other factors to achieve better results. However, these methods perform poorly on image and video DiTs, often leading to content collapse or repetition. RIFLEx (Zhao et al., 2025) mitigates repetition by identifying and attenuating the intrinsic RoPE frequency, yet it still suffers from degraded visual quality. In contrast, our method effectively addresses both content repetition and quality degradation.

**Long Video Generation.** There also exist many approaches to long video generation (Qiu et al., 2023; Wang et al., 2023; Henschel et al., 2025; Kim et al., 2024; Tan et al., 2024; Yin et al., 2025; Wang et al., 2024c; Cai et al., 2025; Li et al., 2025b; Lu et al., 2024; Tan et al., 2025; Jiang et al., 2025a; Gao et al., 2025; Gu et al., 2025), most of which intervene in the diffusion inference process. For instance, FreeNoise (Qiu et al., 2023) enhances temporal consistency via noise initialization, FIFO-Diffusion (Kim et al., 2024) feeds frames sequentially into a denoising window of training length, and Video-Infinity (Tan et al., 2024) exploits distributed computation to scale up video length. While effective for generating long videos, these methods are orthogonal to our length extrapolation strategy, which extends the intrinsic capacity of DiTs to longer sequences and can be readily integrated with them.

In addition to diffusion-based approaches to long video generation, alternative modeling paradigms such as autoregressive methods (Wu et al., 2021; Yan et al., 2021; Hong et al., 2022; Wu et al., 2022; Kondratyuk et al., 2023; Wu et al., 2024; Sun et al., 2024; Wang et al., 2024b) and diffusion forcing (Chen et al., 2024a; Huang et al., 2025; Teng et al., 2025) are also capable of generating long videos. Although our method is designed for diffusion models, it may also offer insights into length extrapolation for these alternative paradigms.

# B More Details of Our Method

## B.1 Derivation of the Periodic Outputs

In this section, we present a formal derivation of Eq. (3). Specifically, the attention score matrix $\boldsymbol{P} \in \mathbb{R}^{L' \times L'}$ satisfies the following properties up to negligible error:

**Prop.1** (Row-wise periodicity): $\boldsymbol{P}_{i,j} = \boldsymbol{P}_{i,j+T}, \forall i \in \{0, \ldots, L'-1\}, j \in \{0, \ldots, L'-T-1\}$, where $T \in \mathbb{N}^+$ corresponds to the observed repetition period in Sec. 3.1.

**Prop.2** (Relative positional invariance): $\boldsymbol{P}_{i,j} = \boldsymbol{P}_{i+p,j+p}, \forall i \in \{0, \ldots, L'-p-1\}, j \in \{0, \ldots, L'-p-1\}$, where $p \in \mathbb{N}^+$ is the relative displacement. In the ffollowing derivation we instantiate $p = T$.

On basis of the above properties, we derive the periodicity of the attention scores and outputs as follows. $\forall i \in \{0, \ldots, L'-T-1\}$,

$$\boldsymbol{O}_{i+T} = \sum_{j=0}^{L'-1} \boldsymbol{P}_{i+T,j} \boldsymbol{V}_j \tag{7}$$

$$= \sum_{j=0}^{L'-T-1} \boldsymbol{P}_{i+T,j} \boldsymbol{V}_j + \sum_{j=L'-T}^{L'-1} \boldsymbol{P}_{i+T,j} \boldsymbol{V}_j \tag{8}$$

$$\overset{\text{Prop.1}}{=} \sum_{j=0}^{L'-T-1} \boldsymbol{P}_{i+T,j+T} \boldsymbol{V}_j + \sum_{j=L'-T}^{L'-1} \boldsymbol{P}_{i+T,j} \boldsymbol{V}_j \tag{9}$$

$$\overset{\text{Prop.2}}{=} \sum_{j=0}^{L'-T-1} \boldsymbol{P}_{i,j} \boldsymbol{V}_j + \sum_{j=L'-T}^{L'-1} \boldsymbol{P}_{i,j-T} \boldsymbol{V}_j \tag{10}$$

$$\overset{\text{Prop.1}}{=} \sum_{j=0}^{L'-T-1} \boldsymbol{P}_{i,j} \boldsymbol{V}_j + \sum_{j=L'-T}^{L'-1} \boldsymbol{P}_{i,j} \boldsymbol{V}_j \tag{11}$$

$$= \sum_{j=0}^{L'-1} \boldsymbol{P}_{i,j} \boldsymbol{V}_j \tag{12}$$

$$= \boldsymbol{O}_i. \tag{13}$$

## B.2 DETAILS OF THE MULTIMODAL ROTARY POSITION EMBEDDING

In this section, we provide the details of the Multimodal RoPE (M-RoPE) (Wang et al., 2024a) introduced in Sec. 2. Specifically, for a token at position $(t, h, w)$, the input vector $\boldsymbol{x} \in \mathbb{R}^D$ is divided into three subspaces of dimensions $d_{\mathcal{T}}, d_{\mathcal{H}}, d_{\mathcal{W}}$, respectively assigned to temporal, height, and width encodings. Each subspace is modulated by its own frequency series $\{\phi_i^{\mathcal{T}}\}_{i=0}^{d_{\mathcal{T}}-1}, \{\phi_i^{\mathcal{H}}\}_{i=d_{\mathcal{T}}}^{d_{\mathcal{T}}+d_{\mathcal{H}}-1}, \{\phi_i^{\mathcal{W}}\}_{i=d_{\mathcal{T}}+d_{\mathcal{H}}}^{D-1}$. Concretely, we define

$$\boldsymbol{f}^{\text{RoPE}}(\boldsymbol{x}, t, h, w)_i = R_i^{\alpha}(p_\alpha) \begin{bmatrix} x_{2i} \\ x_{2i+1} \end{bmatrix}, \quad R_i^{\alpha}(p_\alpha) = \begin{bmatrix} \cos(\phi_i^{\alpha} p_\alpha) & -\sin(\phi_i^{\alpha} p_\alpha) \\ \sin(\phi_i^{\alpha} p_\alpha) & \cos(\phi_i^{\alpha} p_\alpha) \end{bmatrix}, \quad (14)$$

where $\alpha \in \{\mathcal{T}, \mathcal{H}, \mathcal{W}\}$ indexes the temporal, height, and width dimensions with corresponding positions $p_\alpha \in \{t, h, w\}$ and frequency components $\{\phi_i^{\alpha}\}$. The index ranges are

$$i \in \begin{cases} \{0, \ldots, d_{\mathcal{T}}/2 - 1\}, & \alpha = \mathcal{T}, \\ \{d_t/2, \ldots, d_{\mathcal{T}}/2 + d_{\mathcal{H}}/2 - 1\}, & \alpha = \mathcal{H}, \\ \{d_{\mathcal{T}}/2 + d_{\mathcal{H}}/2, \ldots, D/2 - 1\}, & \alpha = \mathcal{W}. \end{cases} \quad (15)$$

After M-RoPE encoding, the queries and keys form $\boldsymbol{Q} \in \mathbb{R}^{L' \times D}$ and $\boldsymbol{K} \in \mathbb{R}^{L' \times D}$. As in Eq. (2), they produce the attention logits matrix $\boldsymbol{S} \in \mathbb{R}^{L' \times L'}$, where the attention logit between the query at $(t, h, w)$, denoted $q_{(t,h,w)}$, and the key at $(t + \Delta t, h + \Delta h, w + \Delta w)$, denoted $k_{(t+\Delta t, h+\Delta h, w+\Delta w)}$, expands explicitly as:

$$\boldsymbol{S}_{(t,h,w),(t+\Delta t,h+\Delta h,w+\Delta w)} = \sum_{i=0}^{d_{\mathcal{T}}/2-1} q_{(t,h,w)}^{(2i:2i+1)\top} \boldsymbol{R}_i^{\mathcal{T}}(\Delta t) k_{(t+\Delta t,h+\Delta h,w+\Delta w)}^{(2i:2i+1)} +$$

$$\sum_{i=d_{\mathcal{T}}/2}^{d_{\mathcal{T}}/2+d_{\mathcal{H}}/2-1} q_{(t,h,w)}^{(2i:2i+1)\top} \boldsymbol{R}_i^{\mathcal{H}}(\Delta h) k_{(t+\Delta t,h+\Delta h,w+\Delta w)}^{(2i:2i+1)} +$$

$$\sum_{i=d_{\mathcal{T}}/2+d_{\mathcal{H}}/2}^{D/2-1} q_{(t,h,w)}^{(2i:2i+1)\top} \boldsymbol{R}_i^{\mathcal{W}}(\Delta w) k_{(t+\Delta t,h+\Delta h,w+\Delta w)}^{(2i:2i+1)} \quad (16)$$

$$= \sum_{i=0}^{d_{\mathcal{T}}/2-1} \left[ \lambda_1^{(i)} \cos(\phi_i^{\mathcal{T}} \Delta t) + \lambda_2^{(i)} \sin(\phi_i^{\mathcal{T}} \Delta t) \right] +$$

$$\sum_{i=d_{\mathcal{T}}/2}^{d_{\mathcal{T}}/2+d_{\mathcal{H}}/2-1} \left[ \lambda_1^{(i)} \cos(\phi_i^{\mathcal{H}} \Delta h) + \lambda_2^{(i)} \sin(\phi_i^{\mathcal{H}} \Delta h) \right] +$$

$$\sum_{i=d_{\mathcal{T}}/2+d_{\mathcal{H}}/2}^{D/2-1} \left[ \lambda_1^{(i)} \cos(\phi_i^{\mathcal{W}} \Delta w) + \lambda_2^{(i)} \sin(\phi_i^{\mathcal{W}} \Delta w) \right], \quad (17)$$

where

$$\lambda_1^{(i)} = q_{(t,h,w)}^{(2i)} k_{(t+\Delta t,h+\Delta h,w+\Delta w)}^{(2i)} + q_{(t,h,w)}^{(2i+1)} k_{(t+\Delta t,h+\Delta h,w+\Delta w)}^{(2i+1)}, \quad (18)$$

$$\lambda_2^{(i)} = q_{(t,h,w)}^{(2i+1)} k_{(t+\Delta t,h+\Delta h,w+\Delta w)}^{(2i)} - q_{(t,h,w)}^{(2i)} k_{(t+\Delta t,h+\Delta h,w+\Delta w)}^{(2i+1)}. \quad (19)$$

## B.3 DERIVATION OF THE STATISTICAL ATTENTION PATTERN $\bar{\boldsymbol{S}}(\Delta t)$

In this section, we present the derivation of Eq. (4) in Sec. 3.2.1. We investigate the row-wise pattern of attention logits by examining the expectation of the attention logits between queries and keys at relative temporal distance $\Delta t$ (i.e., $\mathbb{E}\left[\boldsymbol{S}_{(t,h,w),(t+\Delta t,h,w)}\right]$)[1]. This expectation is taken across attention layers, heads, and query positions. In Appendix B.4, we further show that when the true variance is taken into account, the actual attention logits still follow the same patterns as indicated by this expectation.

---

[1] Strictly speaking, the analysis should target $\boldsymbol{S}_{(t,h,w),(t+\Delta t,h+\Delta h,w+\Delta w)}$ for all $\Delta h, \Delta w$, but as the phenomena are similar across $\Delta h, \Delta w$, we focus on $\boldsymbol{S}_{(t,h,w),(t+\Delta t,h,w)}$ for simplicity.

Specifically, on basis of the formula of M-RoPE (i.e., Eq. (16)), the target expectation is given by[2]

$$
\mathbb{E}_{t,h,w}\Big[\boldsymbol{S}_{(t,h,w),(t+\Delta t,h,w)}\Big] = \mathbb{E}_{t,h,w}\Big[ \sum_{i=0}^{d_\mathcal{T}/2-1} q_{(t,h,w)}^{(2i:2i+1)\top} \boldsymbol{R}_i^\mathcal{T}(\Delta t) k_{(t+\Delta t,h,w)}^{(2i:2i+1)} +
$$

$$
\sum_{i=d_\mathcal{T}/2}^{d_\mathcal{T}/2+d_\mathcal{H}/2-1} q_{(t,h,w)}^{(2i:2i+1)\top} \boldsymbol{R}_i^\mathcal{H}(0) k_{(t+\Delta t,h,w)}^{(2i:2i+1)} + \sum_{i=d_\mathcal{T}/2+d_\mathcal{H}/2}^{D/2-1} q_{(t,h,w)}^{(2i:2i+1)\top} \boldsymbol{R}_i^\mathcal{W}(0) k_{(t+\Delta t,h,w)}^{(2i:2i+1)} \Big] \quad (20)
$$

$$
= \sum_{i=0}^{d_\mathcal{T}/2-1} \Big[ E_1^{(i)} \cos\big(\phi_i^\mathcal{T}\Delta t\big) + E_2^{(i)} \sin\big(\phi_i^\mathcal{T}\Delta t\big) \Big] + \sum_{i=d_\mathcal{T}/2}^{D/2-1} E_1^{(i)}, \quad (21)
$$

where

$$
E_1^{(i)} = \mathbb{E}_{t,h,w}\Big[ q_{(t,h,w)}^{(2i)} k_{(t+\Delta t,h,w)}^{(2i)} + q_{(t,h,w)}^{(2i+1)} k_{(t+\Delta t,h,w)}^{(2i+1)} \Big], \quad (22)
$$

$$
E_2^{(i)} = \mathbb{E}_{t,h,w}\Big[ q_{(t,h,w)}^{(2i+1)} k_{(t+\Delta t,h,w)}^{(2i)} - q_{(t,h,w)}^{(2i)} k_{(t+\Delta t,h,w)}^{(2i+1)} \Big]. \quad (23)
$$

In practice, though the integrands of these expectations are actually functions of $\Delta t$, the empirical statistics in Fig. 9 (col. 1) indicate that their variances with respect to $\Delta t$ are negligible. Hence, we approximate $E_1^{(i)}$ and $E_2^{(i)}$ as constants up to negligible error, which is defined by

$$
E_1^{(i)} \approx \mathbb{E}_{t,h,w,\Delta t}\Big[ q_{(t,h,w)}^{(2i)} k_{(t+\Delta t,h,w)}^{(2i)} + q_{(t,h,w)}^{(2i+1)} k_{(t+\Delta t,h,w)}^{(2i+1)} \Big] =: \hat{E}_1^{(i)}, \quad (24)
$$

$$
E_2^{(i)} \approx \mathbb{E}_{t,h,w,\Delta t}\Big[ q_{(t,h,w)}^{(2i+1)} k_{(t+\Delta t,h,w)}^{(2i)} - q_{(t,h,w)}^{(2i)} k_{(t+\Delta t,h,w)}^{(2i+1)} \Big] =: \hat{E}_2^{(i)}. \quad (25)
$$

By substituting these two expressions into Eq. (22) and Eq. (23), the expected attention logits can be well approximated as $\bar{\boldsymbol{S}}(\Delta t)$, where

$$
\bar{\boldsymbol{S}}(\Delta t) = \sum_{i=0}^{d_\mathcal{T}/2-1} \Big[ \hat{E}_1^{(i)} \cos\big(\phi_i^\mathcal{T}\Delta t\big) + \hat{E}_2^{(i)} \sin\big(\phi_i^\mathcal{T}\Delta t\big) \Big] + \sum_{i=d_\mathcal{T}/2}^{D/2-1} \hat{E}_1^{(i)}. \quad (26)
$$

To simplify the expression, we employ the auxiliary angle formula to rewrite the two trigonometric functions as one, i.e.,

$$
\bar{\boldsymbol{S}}(\Delta t) = \sum_{i=0}^{d_\mathcal{T}/2-1} \Big[ a_i \cos(\phi_i \Delta t + b_i) \Big] + C, \quad (27)
$$

where $a_i = \sqrt{\big[\hat{E}_1^{(i)}\big]^2 + \big[\hat{E}_2^{(i)}\big]^2}, b_i = \mathrm{atan2}(-\hat{E}_2^{(i)}, \hat{E}_1^{(i)})$. Interestingly, as shown in Fig. 9 (col. 2), $\hat{E}_2^{(i)}$ remains consistently close to zero, which in turn makes $b_i$ nearly vanish (for example, $b_0$ is 0.039 for HunyuanVideo). This observation allows us to apply Proposition 1 in Sec. 3.2.1 up to an error of negligible magnitude. Detailed statistical data for $\hat{E}_1^{(i)}, \hat{E}_2^{(i)}, a_i, b_i$ are shown in Fig. 9 (col. 2, 3, 4).

## B.4 CONSISTENCY OF ACTUAL ATTENTION PATTERN WITH $\bar{\boldsymbol{S}}(\Delta t)$

In this section, we investigate the actual attention scores under the true variance, demonstrating that they preserve the same characteristics as the averaged values described in Sec. 3.2.1. As shown in Fig. 10, when the standard deviation over attention layers, heads, and query positions is incorporated into the mean, the attention logits of HunyuanVideo still exhibit clear periodicity at their peaks, whereas those of Wan2.1 remain non-periodic. Therefore, the conclusions drawn in Sec. 3.2.1 from the mean-based analysis hold with strong generality in practice.

---

[2]For brevity, we omit layer and head indices in the expectation notation.

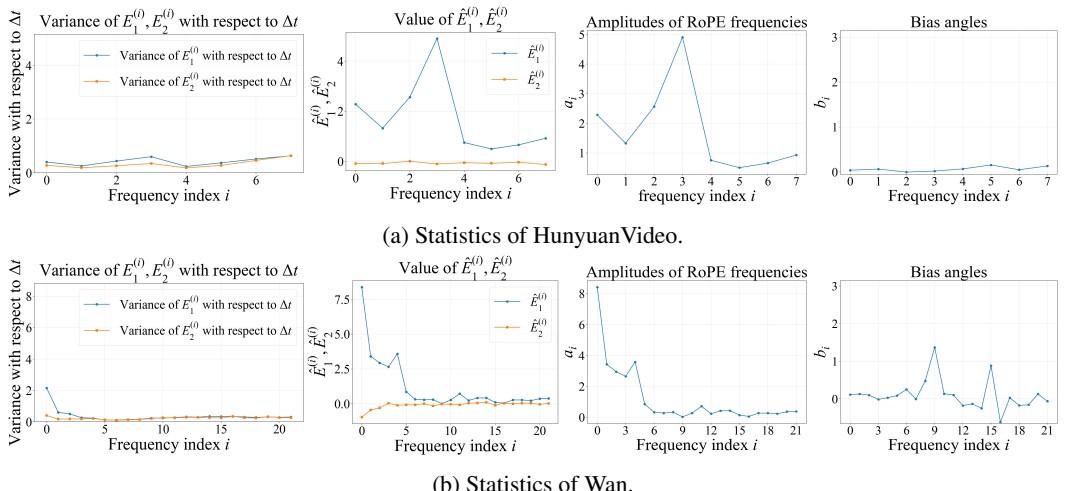

Figure 9: **Statistics of attention logits in HunyuanVideo and Wan.** The variances of $E_1^{(i)}, E_2^{(i)}$ with respect to $\Delta t$ (col. 1) are negligible compared to their expectations (col. 2), making the approximation in Eq. (24), Eq. (25) accurate. The bias angles $b_i$ (col. 4) are close to zero, except for $b_9$ and $b_{15}$ in Wan whose impact is negligible since the corresponding $a_9, a_{15}$ are near zero (col. 3).

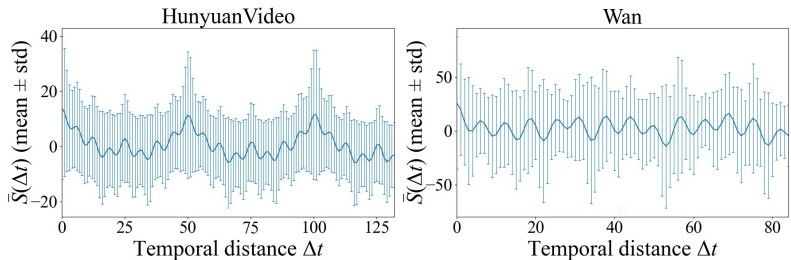

Figure 10: **Attention logits under actual variance.** Even with standard deviation across layers, heads, and query positions, HunyuanVideo retains clear periodic peaks while Wan 2.1 remains non-periodic, confirming the general validity of the mean-based analysis in Sec. 3.2.1.

### B.5   PROOF OF PROPOSITION 1

Proposition 1 is well-known in harmonic analysis and signal processing, and we provide the proof here only for completeness.

*Proof.* **Sufficiency.** If $\phi_i/\phi_{N-1} \in \mathbb{N}^+$ for all $i$, write $\phi_i = k_i\phi_{N-1}$ with $k_i \in \mathbb{N}^+$. Let $T_{N-1} = 2\pi/\phi_{N-1}$. Then for each $i$,

$$\cos\big(\phi_i(\Delta t + T_{N-1})\big) = \cos\big(k_i\phi_{N-1}\Delta t + 2\pi k_i\big) = \cos(\phi_i\Delta t), \quad \forall \Delta t \in \mathbb{R}, \qquad (28)$$

so $f(\Delta t + T_{N-1}) = f(\Delta t)$, $\forall \Delta t \in \mathbb{R}$. Hence $T_{N-1}$ is a period of $f$.

**Necessity.** Suppose $T_{N-1} = 2\pi/\phi_{N-1}$ is a period of $f$. Then for all $\Delta t$,

$$0 = f(\Delta t + T_{N-1}) - f(\Delta t) = \sum_{i=0}^{N-1} a_i \big[\cos(\phi_i\Delta t + \phi_i T_{N-1}) - \cos(\phi_i\Delta t)\big]. \qquad (29)$$

Using $\cos(x + y) - \cos x = (\cos y - 1)\cos x - \sin y \sin x$,

$$0 = \sum_{i=0}^{N-1} a_i \Big[(\cos(\phi_i T_{N-1}) - 1)\cos(\phi_i\Delta t) - \sin(\phi_i T_{N-1})\sin(\phi_i\Delta t)\Big], \quad \forall \Delta t \in \mathbb{R}. \qquad (30)$$

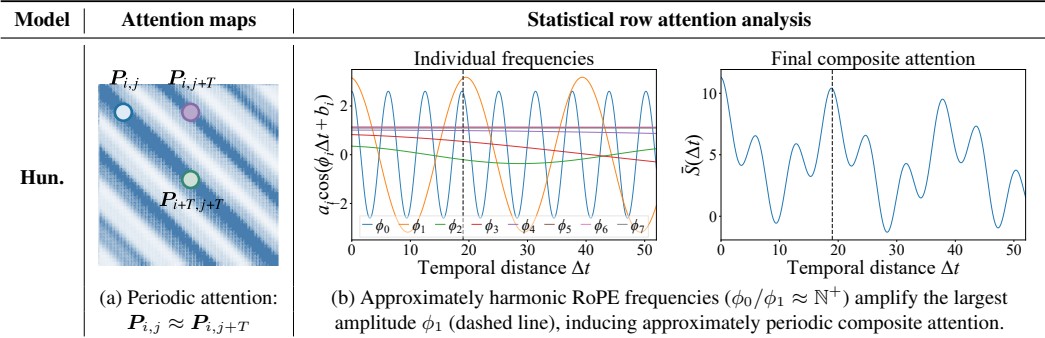

| Model | Attention maps | Statistical row attention analysis |
|---|---|---|

(a) Periodic attention:
$P_{i,j} \approx P_{i,j+T}$

(b) Approximately harmonic RoPE frequencies ($\phi_0/\phi_1 \approx \mathbb{N}^+$) amplify the largest amplitude $\phi_1$ (dashed line), inducing approximately periodic composite attention.

Figure 11: **Periodic attention patterns of CogVideoX.** The RoPE frequencies of CogVideoX approximately satisfy the harmonic condition, which amplifies the largest-amplitude component and thereby induces periodic attention patterns.

The family $\{\cos(\phi_i \cdot), \sin(\phi_i \cdot)\}_i$ with distinct positive $\phi_i$ is linearly independent over $\mathbb{R}$ (e.g., via independence of $e^{\pm i \phi_i t}$). Hence for each $i$,

$$\cos(\phi_i T_{N-1}) - 1 = 0, \qquad \sin(\phi_i T_{N-1}) = 0, \tag{31}$$

so $\phi_i T_{N-1} \in 2\pi \mathbb{Z}$. Substituting $T_{N-1} = 2\pi/\phi_{N-1}$ yields

$$\frac{\phi_i}{\phi_{N-1}} \in \mathbb{N}^+, \tag{32}$$

as all $\phi_i > 0$. $\qquad \square$

### B.6 REMARKS ON PROPOSITION 1

**Relaxed conditions under which the proposition holds approximately.** Although the strict condition for forming harmonics in Proposition 1 is $\phi_i/\phi_{N-1} \in \mathbb{N}^+$, in this section we highlight approximate conditions that can likewise induce a dominant frequency leading to content repetition in videos. Specifically, if $\phi_i/\phi_{N-1}$ is sufficiently close to an integer, constructive amplification can still occur for small $|t|$ (e.g., $|t| \leq 2T_{N-1}$). For example, for CogVideoX, the ratio of the first two frequencies is $\phi_0/\phi_1 = 3.16$, which is close to the integer 3, thereby producing a dominant component that accounts for 50.80% of the total amplitude. This gives rise to an approximately periodic composite attention pattern (Fig. 11), which in turn leads to content repetition (Fig. 13, right).

**Remarks on the strict period of HunyuanVideo.** We herein examine the strict periodicity of HunyuanVideo. Strictly speaking, its fundamental frequency is $\phi_7$, with ratios $\phi_i/\phi_7 = 2^{7-i}, i \in \{0, \ldots, 7\}$. According to Proposition 1, the theoretical period of $\bar{S}(\Delta t)$ is $T_7 = \frac{2\pi}{\phi_7}$. However, as shown in Fig. 9a (col. 3), the amplification contributed by $\phi_7$ is very small, accounting for only 6.677%, which makes its impact negligible. Moreover, its period of 804 is far larger than the extrapolation length (e.g., 132 at $4\times$ extrapolation), rendering the variation of the corresponding component almost imperceptible within this range. The same reasoning applies to $\phi_i$ for $i \in \{4, 5, 6\}$. Consequently, our analysis focuses on $\phi_i$ with $i \in \{0, 1, 2, 3\}$, whose single-frequency contributions are both large enough in amplitude and sufficiently oscillatory to shape $\bar{S}(\Delta t)$.

### B.7 NECESSITY OF CONCENTRATING ON THE TRAINING WINDOW

In this section, we provide detailed experimental evidence supporting the discussion in Sec. 3.2.2 on where sharpened attention focus is most beneficial. Specifically, on Wan with extrapolation ratio $s = 3$, we test four strategies for sharpening attention: concentrating on the leading $\frac{1}{s}$ of each row, the trailing $\frac{1}{s}$, the training window, and the top-$\frac{1}{s}$ tokens according to the original attention scores. As shown in Fig. 12, concentrating on the leading or trailing $\frac{1}{s}$ of each row causes the video to collapse, while top-$\frac{1}{s}$ yields poor visual quality with little dynamics. In contrast, restricting attention to the training window leads to the most significant improvement in video quality.

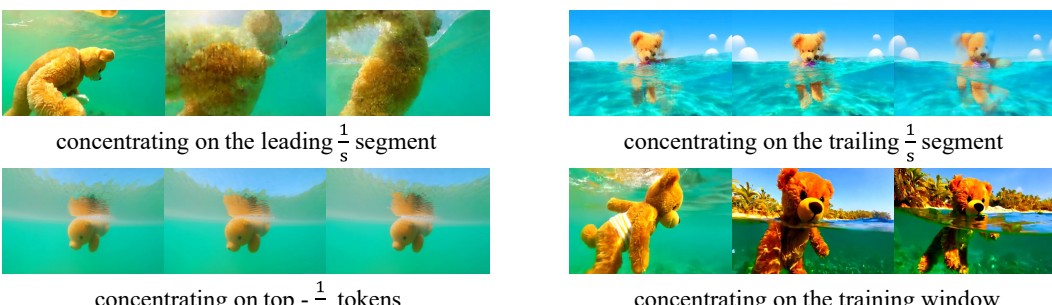

concentrating on the leading $\frac{1}{s}$ segment        concentrating on the trailing $\frac{1}{s}$ segment

concentrating on top - $\frac{1}{s}$ tokens        concentrating on the training window

Figure 12: **Comparison of attention concentration strategies on Wan at** $s = 3$. Concentrating on the leading or trailing $\frac{1}{s}$ of each row collapses the video, and top–$\frac{1}{s}$ yields poor quality with little dynamics. Restricting attention to the training window proves most effective.

## C    MORE DETAILS OF EXPERIMENTS

### C.1    FAILURE MODES OF COGVIDEOX

In this section, we present the manifestation of the failure modes of video length extrapolation as discussed in Sec. 3.1 on an additional model, CogVideoX. As shown in Fig. 13, when extrapolated to three times the normal training length, the generated videos exhibit a sharp decline in both dynamic degree and visual quality, along with noticeable content repetition.

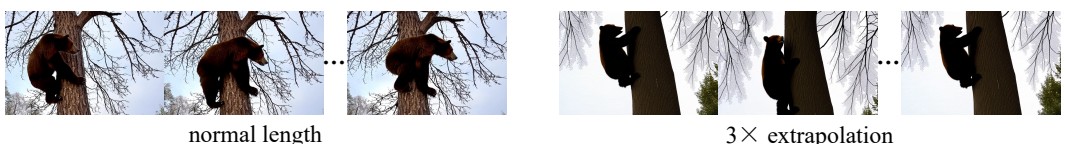

normal length        $3\times$ extrapolation

Figure 13: **Failure modes of CogVideoX under** $3\times$ **extrapolation.** The generated videos show degraded visual quality, reduced dynamics, and clear content repetition, consistent with the failure modes discussed in Sec. 3.1.

### C.2    MORE IMPLEMENTATION DETAILS

In this section, we provide further details of Sec. 4.2.

**The implementation of NoRepeat Score.**    The NoRepeat Score implemented in RIFLEx (Zhao et al., 2025) is only applicable when the content repeats once, which makes it unsuitable for longer extrapolation tasks. We therefore modify it accordingly. Specifically, the computation of the NoRepeat Score consists of two steps: static-video filtering and repeated-frame ratio calculation. In the first step, we uniformly sample 8 frames across the video; if the mean pairwise $L_2$ distance among them falls below a threshold, the video is considered static and discarded. This prevents completely static videos from interfering with subsequent repetition detection. In the second step, we measure the ratio of repeated frames to the total frame count, which defines the NoRepeat Score. Following RIFLEx, we first search around the dominant-frequency period for the frame with the minimal $L_2$ distance to the first frame. This frame is then taken as the start of a candidate repeated sequence. We then compare each frame in this candidate sequence with the corresponding frame at the beginning of the video; frames whose $L_2$ distance is below the threshold are counted as repetitions. Empirically, a threshold of 55 was found to align better with human perception and was consequently applied to both steps. Finally, we report the mean NoRepeat Score across all videos as the final result. The detailed implementation code is included in the supplementary material.

**The implementation of RIFLEx and UltraViCo on Wan.**    Since Wan does not exhibit content repetition, it is not applicable to determine the dominant frequency from the repetition period as per-

formed in Zhao et al. (2025). Instead, following Sec. 3.2.1, we take the largest-amplitude frequency $\phi_0$ as the dominant frequency.

For UltraViCo, the first frame's decay factor is set negative to fix its blurring. We hypothesize that this is caused by the causal design of the video VAE, where the first frame is encoded independently and without temporal compression. As a result, it exhibits different statistical properties from subsequent frames and becomes more sensitive to perturbations.

**Details of the ablation study.** Herein, we detail the setup of the ablation study in Sec. 4.2. Specifically, as shown in Fig. 7 (top), we compare three decay strategies—parabolic, linear, and constant. The parabolic strategy takes the following form:

$$S'_{ij} = \lambda_{ij} \cdot S_{ij}, \quad \text{where} \quad \lambda_{ij} = \begin{cases} 1, & \text{if } |i-j| \le L/2 \text{ or } S_{ij} < 0, \\ \alpha_1(|i-j|/L')^2 + \alpha_2(1-(|i-j|/L')^2), & \text{otherwise}, \end{cases} \tag{33}$$

whereas the linear strategy takes the following form:

$$S'_{ij} = \lambda_{ij} \cdot S_{ij}, \quad \text{where} \quad \lambda_{ij} = \begin{cases} 1, & \text{if } |i-j| \le L/2 \text{ or } S_{ij} < 0, \\ \alpha_1|i-j|/L' + \alpha_2(1-|i-j|/L'), & \text{otherwise}, \end{cases} \tag{34}$$

and the constant strategy is

$$S'_{ij} = \lambda_{ij} \cdot S_{ij}, \quad \text{where} \quad \lambda_{ij} = \begin{cases} 1, & \text{if } |i-j| \le L/2 \text{ or } S_{ij} < 0, \\ \alpha, & \text{otherwise}. \end{cases} \tag{35}$$

We set $\alpha = 0.9$ for the constant strategy, and $\alpha_1 = 0.85, \alpha_2 = 0.95$ for the parabolic and the linear strategies. As shown in Fig. 7 (top), parabolic, linear, and constant decay yield only minor differences, indicating that the key is distinguishing in-window from out-of-window tokens rather than the decay shape.

## C.3 ADDITIONAL EXPERIMENTS OF DIFFERENT EXTRAPOLATION RATIOS AND MODELS

**Settings.** In this section, we provide some additional extrapolation ratios from $s = 2$ to $5$ and models based on 25 prompts from VBench (Huang et al., 2024). To evaluate the generality of UltraViCo, we test $2\times$ extrapolation on HunyuanVideo, Wan, and CogVideoX, as well as $3\times$ and $4\times$ extrapolation on CogVideoX. In addition, we assess $5\times$ extrapolation on HunyuanVideo. For Wan, we set $\alpha = 0.9$. For HunyuanVideo, we use $\gamma = 4$ across all ratios, with $\alpha = 0.95, \beta = 0.6$ at $2\times$ and $\alpha = 0.9, \beta = 0.8$ at $5\times$. For CogVideoX, we use $\gamma = 1$ and $\beta = 0.6$ for all ratios, with $\alpha = 0.9$ at $2\times$ and $3\times$, and $\alpha = 0.85$ at $4\times$. The configurations of other baselines follow Sec. 4.1.

**Results.** We compare UltraViCo with the baselines in Sec. 4.2. As shown in Tab. 3, UltraViCo achieves the best performance across all models and extrapolation ratios, not only avoiding content repetition but also substantially improving video quality. For example, CogVideoX exhibits nearly static videos at $4\times$ extrapolation (Dynamic Degree $\le 16$) with poor visual quality (Imaging Quality $\le 56$), whereas our method significantly enhances both temporal dynamics and visual quality, with Dynamic Degree and Imaging Quality improving by 200% and 13.48%, respectively. Furthermore, at $5\times$ extrapolation, UltraViCo also demonstrates strong performance, surpassing the best baseline scores by 350% in Dynamic Degree and 47.59% in Imaging Quality, indicating the potential of our method to extend to larger extrapolation ratios.

## C.4 MORE QUALITATIVE RESULTS OF OUR METHOD

In this section, we provide additional qualitive results for the experiments in Sec. 4.2. As shown in Fig. 14 and Fig. 15, whether under $3\times$ or $4\times$ extrapolation ratios, and across Wan and CogVideoX, our method consistently achieves substantially superior visual quality and temporal dynamics compared to the baselines. For example, as shown in Fig. 14, the videos generated by various baselines for $3\times$ and $4\times$ extrapolation on Wan are nearly completely static, whereas our method produces highly fluid and natural large-scale motion. Similarly, as shown in Fig. 15, the videos from the baselines are very blurry with dull colors, while our method generates realistic, natural results with rich details.

Table 3: **Quantitative results on VBench for more models and extrapolation**. Note that NoRepeat Score is essentially a binary indicator: red entries indicate visually obvious repetitions, while others show no noticeable repetition.

| Method | Wan with $2\times$ extrapolation | | | | CogVideoX with $3\times$ extrapolation | | | |
|---|---|---|---|---|---|---|---|---|
| | NoRepeat↑ | Dynamic↑ | Quality↑ | Overall↑ | NoRepeat↑ | Dynamic↑ | Quality↑ | Overall↑ |
| PE | N/A | 32 | 58.13 | 23.22 | 82.52 | 16 | 57.91 | 19.59 |
| PI | N/A | 32 | 54.23 | 21.52 | 99.07 | 4 | 54.27 | 18.17 |
| NTK | N/A | 44 | 59.59 | 23.52 | 86.07 | 4 | 55.24 | 19.33 |
| YaRN | N/A | 24 | 55.14 | 21.57 | 97.47 | 0 | 53.96 | 18.05 |
| TASR | N/A | 36 | 59.97 | 23.70 | 97.93 | 8 | 55.75 | 19.24 |
| RIFLEx | N/A | 16 | 48.15 | 20.34 | 97.86 | 8 | 55.31 | 19.03 |
| **Ours** | N/A | **68** | **66.88** | **25.28** | 99.38 | **32** | **60.09** | **24.77** |

| Method | HunyuanVideo with $2\times$ extrapolation | | | | CogVideoX with $4\times$ extrapolation | | | |
|---|---|---|---|---|---|---|---|---|
| | NoRepeat↑ | Dynamic↑ | Quality↑ | Overall↑ | NoRepeat↑ | Dynamic↑ | Quality↑ | Overall↑ |
| PE | 80.43 | 40 | 62.67 | 24.36 | 76.57 | 16 | 55.25 | 17.27 |
| PI | 98.87 | 4 | 52.35 | 23.55 | 88.53 | 4 | 46.82 | 16.63 |
| NTK | 94.97 | 32 | 65.47 | 24.62 | 78.89 | 2 | 52.74 | 18.14 |
| YaRN | 97.99 | 4 | 52.87 | 23.26 | 94.75 | 4 | 47.36 | 16.90 |
| TASR | 94.85 | 36 | 64.55 | 24.59 | 99.13 | 16 | 46.75 | 17.28 |
| RIFLEx | 97.27 | 36 | 65.19 | 24.52 | 97.00 | 12 | 50.59 | 16.66 |
| Ours | 97.53 | **44** | **66.50** | **24.82** | 96.79 | **48** | **62.70** | **25.39** |

| Method | CogVideoX with $2\times$ extrapolation | | | | HunyuanVideo with $5\times$ extrapolation | | | |
|---|---|---|---|---|---|---|---|---|
| | NoRepeat↑ | Dynamic↑ | Quality↑ | Overall↑ | NoRepeat↑ | Dynamic↑ | Quality↑ | Overall↑ |
| PE | 92.31 | 28 | 64.28 | 22.83 | 30.78 | 4 | 39.04 | 15.64 |
| PI | 98.85 | 8 | 57.11 | 21.88 | 81.58 | 0 | 36.63 | 16.76 |
| NTK | 94.66 | 16 | 63.04 | 23.55 | 71.54 | 8 | 43.43 | 17.78 |
| YaRN | 98.81 | 8 | 58.83 | 21.81 | 77.70 | 0 | 37.88 | 17.85 |
| TASR | 95.91 | 16 | 62.17 | 23.44 | 35.31 | 8 | 42.88 | 17.88 |
| RIFLEx | 99.42 | 16 | 60.30 | 23.28 | 53.65 | 4 | 40.55 | 15.71 |
| **Ours** | 98.92 | **32** | **64.39** | **25.36** | 99.44 | **36** | **64.10** | **24.16** |

Moreover, we present another downstream task in Fig. 16, where generation is performed based on a given pose. Our method achieves high quality and dynamic results while closely following the given conditions.

## C.5 ACCELERATION OF ULTRAVICO VIA SPARSE ATTENTION AND DISTILLATION

Building upon recent advances in sparse-attention-based video acceleration and distillation (Team, 2024b), UltraViCo achieves about $16\times$ speed-up without compromising performance (see Table 7).

## C.6 RUNTIME AND MEMORY COST

As shown in Table 8, built on top of FlashAttention (Dao et al., 2022) and SageAttention (Zhang et al., 2024b;a), UltraViCo incurs almost no additional overhead in either latency or memory usage.

## D FURTHER DETAILS OF ULTRAVICO

### D.1 ULTRAVICO WITH EFFIEIENT ONLINE ATTENTION

UltraViCo does not require materializing the full attention matrix and can be seamlessly integrated into efficient online attention kernels. Herein, we present its implementation based on FlashAttention, as illustrated by Algorithm 1.

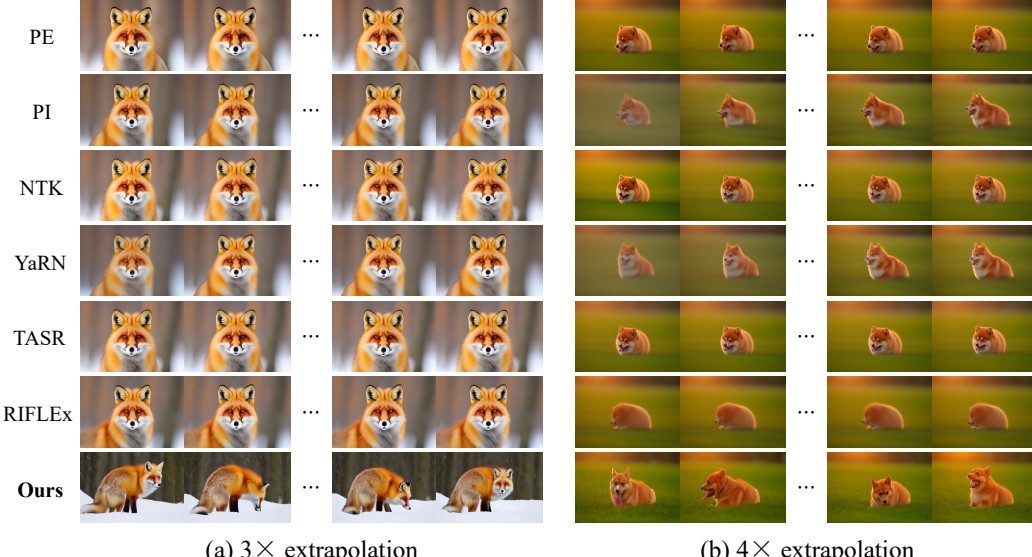

(a) 3× extrapolation

(b) 4× extrapolation

Figure 14: **Qualitative results on Wan**. The baselines produce nearly static videos with poor visual quality, whereas our method achieves significantly better quality and much more motion.

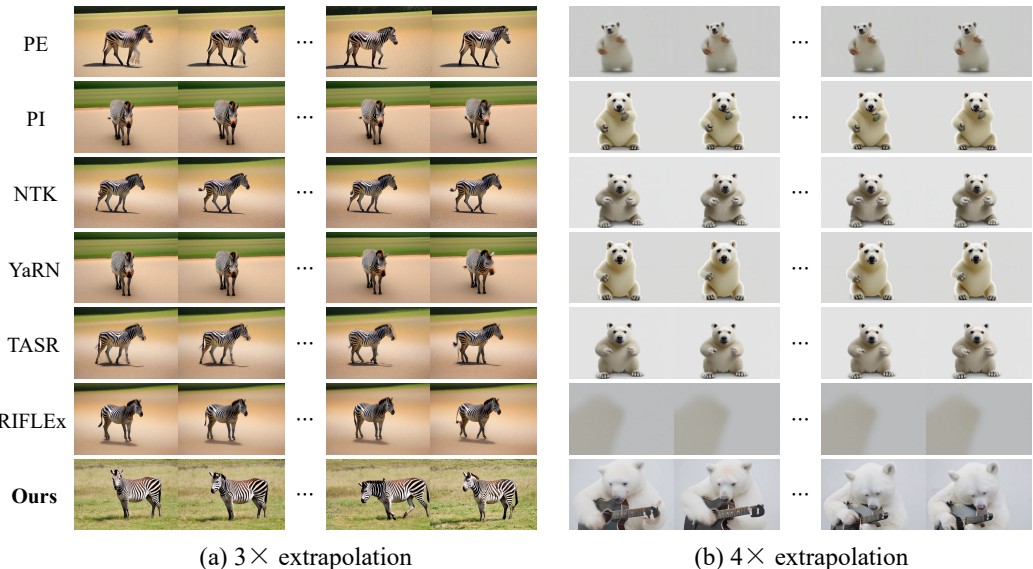

(a) 3× extrapolation

(b) 4× extrapolation

Figure 15: **Qualitative results on CogVideoX**. The baselines produce nearly static videos with poor visual quality, whereas our method generates realistic results with rich details and fluid motion.

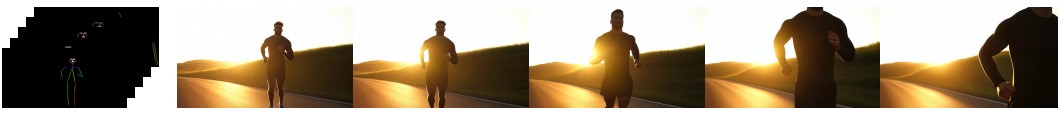

Figure 16: **Our method for pose-guided video generation**. Our method closely aligns with the given pose conditions, while ensuring high dynamic range and excellent visual quality.

---

**Algorithm 1** UltraViCo FlashAttention Kernel

---

**Require:** Matrices $Q, K, V \in \mathbb{R}^{N \times d}$, block size $b_q, b_{kv}$.
1: Divide $Q$ into $T_m = N/b_q$ blocks $\{Q_m\}$, and divide $K, V$ into $T_n = N/b_{kv}$ blocks $\{K_n\}$ and $\{V_n\}$;
2: **for m** in $[1, T_m]$ **do**
3:     **for n** in $[1, T_n]$ **do**
4:         $\vec{i} = m \times b_q + \mathrm{range}(0, b_q),\ \vec{j} = n \times b_{kv} + \mathrm{range}(0, b_{kv}),\ \vec{i} \in \mathbb{R}^{1 \times b_q}, \vec{j} \in \mathbb{R}^{1 \times b_{kv}}$ ;
5:         Initialize $\lambda \in \mathbb{R}^{b_q \times b_{kv}}$ to 0 ;
6:         $\lambda = $ Eq. $6(\vec{i}, \vec{j})$ ;
7:         $S_m^n = \lambda Q_m K_n^T$ ;
8:         $p_m^n = \max(p_m^{n-1}, \mathrm{rowmax}(S_m^n))$ ;
9:         $\widetilde{P}_m^n = \exp(S_m^n - p_m^n)$ ;
10:        $l_m^n = e^{p_m^{n-1} - p_m^n} l_m^{n-1} + \mathrm{rowsum}(\widetilde{P}_i^j)$ ;
11:        $O_m^n = \mathrm{diag}(e^{p_m^{n-1} - p_m^n}) O_m^{n-1} + \widetilde{P}_m^n V_n$ ;
12:     **end for**
13:     $O_m = \mathrm{diag}(l_m^{T_n})^{-1} O_m^{T_n}$ ;
14: **end for**
15: **return** $O = \{O_m\}$;

---

## D.2 ABLATION ON HYPERPARAMETERS

In this section, we present more detailed illustrative ablation results for the hyperparameters $\alpha$ and $\beta$. The detailed sensitivity curve is shown in Fig. 17, while the illustrative ablations on the independent effects of $\alpha$ and $\beta$ in the main experiments are reported in Tab. 6.

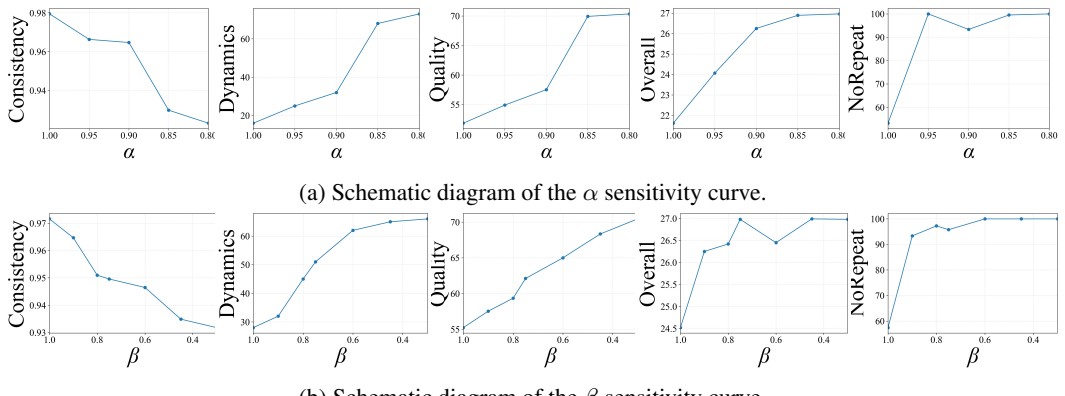

(a) Schematic diagram of the $\alpha$ sensitivity curve.

(b) Schematic diagram of the $\beta$ sensitivity curve.

Figure 17: **Illustration of the hyperparameter sensitivity curve.**

Table 4: **Illustrative sensitivity analysis of $\alpha$ on Hunyuan at $3\times$ extrapolation. We set $\beta$ equal to $\alpha$, i.e., a single decay factor is shared globally.**

| $\alpha$ | Consistency↑ | Dynamics↑ | Quality↑ | Overall↑ | NoRepeat↑ |
|---|---|---|---|---|---|
| 1.0 | 0.9795 | 16 | 51.85 | 21.62 | 53.17 |
| 0.95 | 0.9663 | 25 | 54.92 | 24.07 | 100 |
| 0.9 | 0.9647 | 32 | 57.53 | 26.25 | 93.34 |
| 0.85 | 0.9298 | 68 | 69.93 | 26.89 | 99.53 |
| 0.8 | 0.9231 | 73 | 70.35 | 26.96 | 100 |

Table 5: **Illustrative sensitivity analysis of $\beta$ on Hunyuan at $3\times$ extrapolation. We set $\alpha = 0.9$ across all settings.**

| $\beta$ | Consistency↑ | Dynamics↑ | Quality↑ | Overall↑ | NoRepeat↑ |
|---|---|---|---|---|---|
| 1.0 | 0.9716 | 28 | 55.23 | 24.52 | 57.42 |
| 0.9 | 0.9647 | 32 | 57.53 | 26.25 | 93.34 |
| 0.8 | 0.9510 | 45 | 59.35 | 26.42 | 97.25 |
| 0.75 | 0.9496 | 51 | 62.11 | 26.98 | 95.77 |
| 0.6 | 0.9465 | 62 | 65.00 | 26.45 | 100 |
| 0.45 | 0.9349 | 65 | 68.34 | 26.99 | 100 |
| 0.3 | 0.9318 | 66 | 70.45 | 26.98 | 100 |

Table 6: **Illustrative ablation experiments that independently examine the individual effects of $\alpha$ and $\beta$.**

| Method | Consistency↑ | Dynamics↑ | Quality↑ | Overall↑ | NoRepeat↑ |
|---|---|---|---|---|---|
| HunyuanVideo with $3\times$ extrapolation | | | | | |
| $\alpha = 1, \beta = 1$ | 0.9795 | 16 | 51.85 | 21.62 | 53.17 |
| $\alpha = 0.9, \beta = 1$ | 0.9716 | 28 | 55.23 | 24.52 | 57.42 |
| $\alpha = 1, \beta = 0.6$ | 0.9784 | 25 | 55.13 | 23.13 | 93.52 |
| $\alpha = 0.9, \beta = 0.6$ | 0.9465 | 62 | 65.00 | 26.45 | 100 |
| Wan2.1-1.3B with $3\times$ extrapolation | | | | | |
| $\alpha = 1$ | 0.9419 | 6 | 56.28 | 18.53 | – |
| $\alpha = 0.9$ | 0.9444 | 46 | 62.43 | 23.21 | – |

Table 7: **Illustrative performance when combined with recent video-acceleration methods on HunyuanVideo.**

| Setting | Time Cost↓ | Consistency↑ | Dynamics↑ | Quality↑ | Overall↑ | NoRepeat↑ |
|---|---|---|---|---|---|---|
| $3\times$ | 5 GPU·hours | 0.9465 | 62 | 65.00 | 26.45 | 100 |
| $3\times$ with FastVideo | 0.3 GPU·hours | 0.9432 | 64 | 63.89 | 25.98 | 100 |
| $4\times$ | 8 GPU·hours | 0.9491 | 42 | 66.54 | 24.52 | 99.87 |
| $4\times$ with FastVideo | 0.5 GPU·hours | 0.9399 | 40 | 62.24 | 24.83 | 96.32 |

Table 8: **Illustrative runtime and memory comparison.** Note that SageAttention is optimized for 4090-like architectures; on A800, its runtime is comparable to FlashAttention.

| Model / Method | Time (s / iter) | Memory (per GPU) |
|---|---|---|
| HunyuanVideo ($3\times$ extrapolation) | | |
| SageAttention | 341.2 | 73188M |
| SageAttention + Ours | 349.6 | 72346M |
| FlashAttention | 349.3 | 76030M |
| FlashAttention + Ours | 355.3 | 75932M |
| Wan ($3\times$ extrapolation) | | |
| SageAttention | 32.13 | 24342M |
| SageAttention + Ours | 34.12 | 24342M |
| FlashAttention | 32.64 | 24349M |
| FlashAttention + Ours | 33.74 | 24346M |

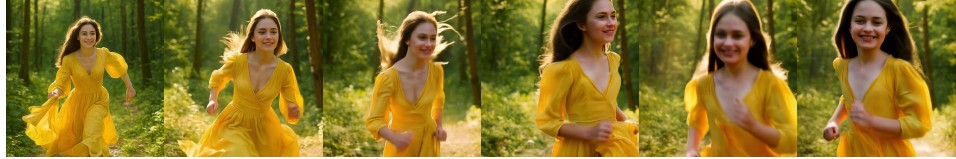

(a) Performance of the video-continuation baseline alone.

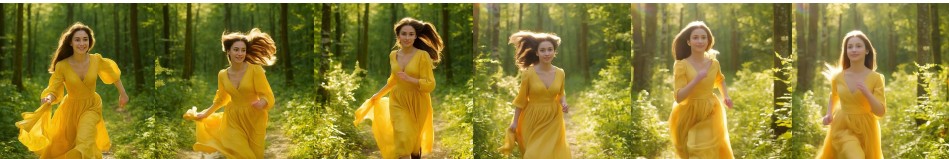

(b) Illustration of combining UltraViCo with the video-continuation method.

Figure 18: **Application of UltraViCo to segment-wise long-video generation.** (a) Wan2.2-TI2V uses only a few ending frames, causing identity drift; (b) UltraViCo alleviates this issue.

