# OpenReview forum: "UltraViCo: Breaking Extrapolation Limits in Video Diffusion Transformers"
_ICLR.cc/2026/Conference — ICLR 2026 Poster_

### Official Review · Reviewer_3Ey3 · 2025-10-20

**Soundness:** 3
**Presentation:** 2
**Contribution:** 2
**Rating:** 4
**Confidence:** 3

**Summary:**

Video Length Extrapolation adapts pretrained video diffusion transformers to generate sequences longer than training in a single pass, but models typically loop content and lose quality beyond trained length 𝐿. The paper attributes both failures to attention dispersion. UltraViCo, a training-free method that decays attention to tokens beyond the training window, restores focus, extending extrapolation from 2× to 4×. This outperforms baselines across models and extrapolation ratios.

**Strengths:**

- Training-free, plug-and-play: Requires no additional data or fine-tuning.

- Repetition mitigation via RoPE insight: Identifies the correlation between attention periodicity observed across models and RoPE-induced harmonics, effectively reducing repetition during video length extrapolation.

- Quality preservation through attention concentration: Alleviates attention dispersion beyond the training window, mitigating quality degradation in long-horizon generation across diverse video generation models.

**Weaknesses:**

1. Non-uniform evaluation prompts: Prompts are not standardized across models (e.g., UltraViCo uses 100 prompts, HunYuan uses 25), complicating fairness and direct comparability.

2. Limited analysis of Eq. (6): The paper provides insufficient sensitivity/ablation on how changes in 𝛼 and 𝛽 affect outcomes, leaving robustness and trade-offs unclear.

3. Hyperparameter dependence: 𝛼 and 𝛽 must be tuned per model and per extrapolation ratio, reducing generality and increasing deployment friction—hindering active, broad use despite the training-free claim.

4. Minor suggestions: The notation in Eq. (6) makes it unclear whether the constant 1 or 𝛽 has precedence (i.e., which term is intended to dominate or be applied first). It would be clearer to revise the expression to make the precedence explicit.

**Questions:**

1. Given existing approaches for long video generation that rely on inference-time modifications (e.g., FIFO), what is the specific motivation and unique significance of studying video length extrapolation as a separate paradigm?

2. In Figure 3, when visualizing Eq. (4), how many video samples were used? Was the visualization averaged over multiple videos, or based on a single example? Because Eq. (4) involves both queries and keys, it seems necessary to report statistics over multiple videos to confirm consistency of the observations.

3. For Eq. (6), is there an analysis of how varying 𝛼 and 𝛽 affects the results (e.g., sensitivity curves or robustness across ranges)?

4. Regarding the decay factors introduced in Eq. (6), could you include an ablation with quantitative metrics in Table 1 to show their individual contributions?

---

> ### Author Response · Authors · 2025-11-21
> **Author Response to Reviewer 3Ey3**
>
> We sincerely thank Reviewer 3Ey3 for the valuable suggestions. We have thoroughly addressed the detailed comments as follows.
>
> ## Q1. Non-uniform evaluation prompts: Prompts are not standardized across models (e.g., UltraViCo uses 100 prompts, HunYuan uses 25), complicating fairness and direct comparability.
>
> We clarify that UltraViCo is applied on top of each base model (e.g., Hunyuan) to extend its effective window; it is not compared against the base model itself. In Table 1, although different base model settings may use different numbers of prompts, **within each setting UltraViCo and all baselines are evaluated on the same base model using the same prompt set, ensuring fair comparison**.
>
> As suggested, we additionally evaluated all methods on the Hunyuan 4× extrapolation setting using 100 prompts to fully align the evaluation across all settings in **Rebuttal Table 7**, and the results consistently show that UltraViCo outperforms all baselines. We have updated the metric in Table 1 of the revised paper.
>
> **Rebuttal Table 7**:  4× extrapolation results on Hunyuan using 100 unified prompts for alignment across all settings.
>
> | Method | Dynamics $\uparrow$ | Quality $\uparrow$ | Overall $\uparrow$ | NoRepeat $\uparrow$ |
> |--------|-------------|------------|-------------|-------------|
> | PE     | 14          | 47.12      | 17.61       | 31.41       |
> | PI     | 0           | 42.19      | 17.83       | 70.93       |
> | NTK    | 10          | 50.01      | 18.92       | 72.39       |
> | YaRN   | 1           | 41.37      | 18.53       | 62.87       |
> | TASR   | 14          | 46.81      | 18.47       | 51.28       |
> | RIFLEx | 11          | 41.02      | 16.47       | 52.84       |
> | **Ours**  | **42**      | **66.54**   | **24.52**    | **99.87**    |
>
>
>
>
>
> ## Q2. Limited analysis of Eq. (6): Insufficient sensitivity/ablation on $\alpha$ and $\beta$ . Hyperparameter dependence: 𝛼 and 𝛽 must be tuned per model and per extrapolation ratio, reducing generality and increasing deployment friction—hindering active, broad use despite the training-free claim.
>
> Thank you for this insightful suggestion. As suggested ( see **Common Concern 2**), we add a detailed analysis of $\alpha$ and $\beta$ and the results shows a consistent and robust pattern across models and extrapolation ratios: when $\alpha$ ≥ 0.9 and $\beta$ ≥ 0.6, visual quality and motion dynamics improve substantially, while temporal consistency remains comparable to the baseline. Therefore, we recommend $\alpha$ = 0.9 and $\beta$ = 0.6 as default settings, which already outperform the baseline. For users who prefer slightly stronger motion or stronger consistency, a light adjustment around the defalt is enough. Overall, the method remains training-free and requires minimal tuning in practice.  We have incorporated the above discussion into the Experiments section of the revised paper.
>
> ## Q3. Minor suggestions: The notation in Eq. (6) makes it unclear whether the constant 1 or 𝛽 has precedence (i.e., which term is intended to dominate or be applied first). It would be clearer to revise the expression to make the precedence explicit.
>
>
> Thanks for the reviewer’s careful observation. As suggested, we have revised Eq. (6) by adding explicit parentheses around the $\beta$ -dependent term, making it clear that the constant 1 is applied first and removing any ambiguity regarding precedence.  This correction has been incorporated into Eq. (6) of the revised paper.
>
>
>
> ## Q4. Given existing approaches for long video generation that rely on inference-time modifications (e.g., FIFO), what is the specific motivation and unique significance of studying video length extrapolation as a separate paradigm?
>
> As discussed in **Common Concern 1**, our work focuses on directly extending the effective training window of any video diffusion transformer, which is fundamentally orthogonal to inference-time long-video approaches such as FIFO. UltraViCo can be seamlessly combined with these methods to further enhance long-term temporal consistency, highlighting the unique significance of studying video length extrapolation as an independent paradigm. We have incorporated the above discussion into the Experiments section of the revised paper.

---

> > ### Author Response · Authors · 2025-11-21
> >
> > ## Q5. In Figure 3, when visualizing Eq. (4), how many video samples were used? Was the visualization averaged over multiple videos, or based on a single example? Because Eq. (4) involves both queries and keys, it seems necessary to report statistics over multiple videos to confirm consistency of the observations.
> >
> > We used 10 videos for this visualization. Empirically, we found that different prompts (videos) and different denoising time steps have only minor influence on the attention patterns. The structure of each layer/head’s pattern is predominantly determined by the model itself, which is consistent with the observations reported in STA [1].
> >
> > As shown in Figure 3 in the revised paper, we add standard deviation bars across videos to explicitly illustrate this consistency. We thank the reviewer for this insightful suggestion, which has significantly improved the clarity and reliability of our analysis.
> >
> > [1] Fast Video Generation with SLIDING TILE ATTENTION
> >
> > ## Q6. Regarding the decay factors introduced in Eq. (6), could you include an ablation with quantitative metrics in Table 1 to show their individual contributions?
> >
> > As suggested, we add an ablation with quantitative metrics in **Rebuttal Table 8**. Notably, the Wan model involves only the hyperparameter $\alpha$, since it does not include the mid-band region where repetition risk arises.
> >
> > For the content repetition (measured by NoRepeat Score), the decay factor $\beta$ plays the dominant role: a smaller $\beta$ more strongly suppresses the periodic attention leakage described in Proposition 1, leading to noticeably higher NoRepeat scores. For visual quality, both $\alpha$ and $\beta$ provide consistent benefits, likely because both improve the model’s ability to maintain concentrated attention within the effective training window. We have incorporated the results from Rebuttal Table 8 into Table 6 of the revised paper.
> >
> >
> > **Rebuttal Table 8.** Ablation results of individual contributions.
> > | Method | Consistency $\uparrow$ | Dynamics $\uparrow$ | Quality $\uparrow$ | Overall $\uparrow$ | NoRepeat $\uparrow$ |
> > |--------|----------------|-------------|------------|-------------|-------------|
> > |HunyuanVideo with 3× extrapolation |
> > |$\alpha=1,\beta=1$|0.9795   | 16       | 51.85         | 21.62       | 53.17       |
> > |$\alpha=0.9, \beta=1$    | 0.9716     | 28      | 55.23   | 24.52   | 57.42              |
> > |$\alpha=1, \beta=0.6$    |0.9784  | 25     |55.13|23.13    | 93.52  |
> > |$\alpha=0.9, \beta=0.6$    |0.9465   | 62     |65.00   |26.45     | 100   |
> > |Wan with 3× extrapolation|
> > |$\alpha=1$  | 0.9419        | 6          | 56.28      | 18.53      | --       |
> > |$\alpha=0.9$    | 0.9444        | 46        | 62.43     | 23.21      | --       |

---

> ### Comment · Reviewer_3Ey3 · 2025-11-26
>
> Thank you for addressing my concerns.
> I have reviewed your response, and I will adjust my rating accordingly.

---

> > ### Author Response · Authors · 2025-11-26
> > **Thanks for the feedback**
> >
> > Thank you for the appreciation of our response and the update on the score. We highly appreciate it.

---

### Official Review · Reviewer_Tf16 · 2025-10-30

**Soundness:** 3
**Presentation:** 3
**Contribution:** 3
**Rating:** 6
**Confidence:** 4

**Summary:**

This paper tackles video length extrapolation for generating videos longer than the training sequence length in diffusion-based video transformers. State-of-the-art text-to-video diffusion models are trained on fixed-duration clips (e.g. 5 seconds) and struggle beyond that length. The paper highlights the two failure modes that emerge as sequence length grows: (i) periodic content repetition, and (ii) quality degradation affecting all models. These issues worsen at longer extrapolations, limiting practical use. Prior work such as RIFLEx addressed repetition via positional encoding tweaks but did not solve the widespread quality drop. The author(s) of this paper identify a unified cause behind both failures, i.e, attention dispersion. When generating beyond the trained window, new frames dilute the model's learned attention focus, causing it to spread attention too broadly. In extreme cases, Rotary Positional Embedding (RoPE) harmonics can organize this dispersion into a periodic pattern, explaining the looping behavior. More generally, dispersion leads to the model losing focus on fine details and mixing unrelated temporal contexts. Building on this insight, the paper introduces UltraViCo. It is a training-free, plug-and-play method to restore the model’s focus during long video generation. The author(s) claim impressive gains in video quality and temporal dynamics on multiple text-to-video diffusion models (HunyuanVideo 1.3B, CogVideoX 5B, and Wan 2.1) and various extrapolation lengths.

**Strengths:**

1. The paper provides a unification of two major failure modes under the single concept of attention dispersion. To me, this is a good contribution contending the earlier positional encoding-centric explanations.

2. The approach presented in the paper requires no additional training or model modifications, making it a plug-and-play inference-time fix. Despite its simplicity, it tackles both identified issues simultaneously.

3. The paper gives importance to practical implementation. Integrating the attention decay into a custom FlashAttention kernel shows strong engineering effort so that the method works on large-scale models without running out of memory.

**Weaknesses:**

1. UltraViCo down-weights the influence of very distant frames. A possible side effect is that the model might lose some long-term context or consistency. The paper asserts that important content is preserved while removing irrelevant far-context influence but it does not rigorously quantify scene consistency over extremely long videos. In one ablation, using too small an $\alpha$ caused a car tire to disappear in later frames, indicating that overly concentrating attention can indeed harm persistent content.

2.  The approach requires a massive computational cost for generating long videos. The authors report needing ~8 A800 GPU-hours per video for 4x extrapolation on the largest model, which forced them to use a reduced number of samples for evaluation.

3. The paper focuses comparisons on inference-time extrapolation techniques (RoPE/PE variants and RIFLEx), which is appropriate, yet omits direct comparison to some alternative approaches for long video generation. For example, sequential generation methods like FIFO-Diffusion or diffusion re-initialization schemes (NUWA-XL, Flexible Diffusion Modeling for long videos Neurips 2022, interactive video generation via domain adaptation, and others) could also produce long videos. The authors cite these works and justifiably note they are different.

4. The theoretical analysis stops short of fully justifying the chosen solution. The idea of suppressing out-of-window attention is intuitively motivated and empirically validated, but there isn’t a formal framework proving it as an optimal or necessary fix. For example, the harmonic analysis (Proposition 1) explains when periodic repetition arises, but there’s no analogous theory for why a constant decay factor is the best way to concentrate attention.

**Questions:**

1. Does suppressing distant-frame attention risk any long-term coherence issues? In particular, how does UltraViCo ensure that important elements present in the first few frames like background setting, main character identity, etc. are still represented in later frames, given those early frames’ influence is decayed?

2. The paper positions as complementary to other long-video generation techniques. Have the author(s) tested or considered combining UltraViCo with such methods? For instance, could one apply UltraViCo in a sliding-window generation to further improve consistency at the window boundaries, or use it alongside a method like FreeNoise to additionally boost temporal stability?

3. The implementation mentions setting the first frame decay factor to a negative value to counteract blurring. Could the authors clarify why the first frame needed special treatment?

4. How should one choose the decay parameters $(\alpha, \beta, \gamma)$ for a new model or a different extrapolation ratio? The results show different optimal values for different scenarios (for e.g.  stronger decay for 5x on HunyuanVideo, $\alpha = 0.85$ for CogVideoX at 4x).

---

> ### Author Response · Authors · 2025-11-21
> **Author Response to Reviewer Tf16**
>
> We sincerely thank Reviewer Tf16 for the recognition of our work. The further questions are addressed as follows.
>
> ## Q1. UltraViCo down-weights the influence of very distant frames, and might lose some long-term consistency. The paper does not quantify scene consistency over extremely long videos.
>
>
> Thanks for the insightful suggestion. As recommended, we add background consistency (Consistency) as an additional scene-consistency metric in **Rebuttal Table 5**, together with a detailed analysis of the roles of $\alpha$ and $\beta$ in **Common Concern 2**. Empirically, we find that an appropriate choice of $\alpha$ and $\beta$ substantially improves visual quality and motion dynamics while maintaining comparable temporal consistency. For instance, on Wan with 4× resolution extrapolation, our method increases the dynamics score by over 400% (11 → 47), yet consistency remains stable (0.9415 → 0.9484). Notably, dynamics and consistency naturally trade off and should be considered jointly. On HunyuanVideo, although our consistency metric decreases slightly, it remains around 0.95, which is visually stable (e.g., Wan’s training-horizon consistency is ≈0.95). Moreover, the baselines achieve high consistency because their motion dynamics are extremely low; such consistency is therefore less meaningful.
>
>
> Overall, UltraViCo can improve visual quality and motion dynamics while maintaining comparable temporal consistency and the introduced $\alpha$/$\beta$ controls provide a flexible balance between motion expressiveness and temporal stability. We have incorporated the consistency metric into Table 1 of the revised paper.
>
> **Rebuttal Table 5.** Quantative results. 4× extrapolation on HunyuanVideo using 100 unified prompts to ensure alignment across all settings, as suggested in Q2.
> | Method | Consistency $\uparrow$ | Dynamics $\uparrow$ | Quality $\uparrow$ | Overall $\uparrow$ | NoRepeat $\uparrow$ |
> |--------|----------------|-------------|------------|-------------|-------------|
> |Wan|
> | Training length     | 0.9554     | 51         | 70.34  | 24.25      | --      |
> |PE with 3× extrapolation    | 0.9419        | 6          | 56.28      | 18.53      | --       |
> |Ours with 3× extrapolation ($\alpha=0.9$)    | **0.9444**        | **46**       | **62.43**     | **23.21**      | --       |
> |PE with 4× extrapolation    | 0.9415         | 11         | 55.25       | 16.65       | --       |
> |Ours with 4× extrapolation ($\alpha=0.9$)    | **0.9484**        | **47**       | **59.36**     | **21.61**      | --       |
> |HunyuanVideo|
> | Training length     | 0.9786         | 71          |69.31      |  26.81     | --       | --      |
> |PE with 3× extrapolation |**0.9795**   | 16       | 51.85         | 21.62       | 53.17       |
> |Ours with 3× extrapolation ($\alpha=0.9, \beta=0.6$)    |0.9465   | **62**     |**65.00**   |**26.45**     | **100**   |
> |Ours with 3× extrapolation ($\alpha=0.9, \beta=0.9$)    | 0.9647        | 32       | 57.53    | 26.25     | 93.34       |
> |PE with 4× extrapolation    | **0.9802**        | 14         | 47.12      | 17.61       | 31.41      |
> |Ours with 4× extrapolation ($\alpha=0.9, \beta=0.8$)    | 0.9491       | **42**      | **66.54**    | **24.52**    | **99.87**     |
>
>
>
>
> ## Q2. The approach requires a massive computational cost for generating long videos. The authors report needing ~8 A800 GPU-hours per video for 4x extrapolation on the largest model, which forced them to use a reduced number of samples for evaluation.
>
> We acknowledge that the high computational cost is a common limitation of full-attention video diffusion models when generating long videos, rather than a constraint specific to our method. Fortunately, as shown in the **Rebuttal Table 6**, building upon recent advances in sparse-attention–based video acceleration and distillation [1], UltraViCo can achieve 16x without harming performance. Notably, to address the reviewer’s concern regarding sample size, all methods under the Hunyuan 4× extrapolation setting in **Rebuttal Table 6** are evaluated using 100 unified prompts, ensuring fully aligned comparisons across all settings.
>
> [1] FastVideo: A Unified Framework for Accelerated Video Generation
>
> **Rebuttal Table 6.** Performance when combined with recent video-acceleration methods on HunyuanVideo.
> | Setting |Time Cost $\downarrow$| Consistency $\uparrow$ | Dynamics $\uparrow$ | Quality $\uparrow$ | Overall $\uparrow$ | NoRepeat $\uparrow$ |
> |--------|----------------|----------------|-------------|------------|-------------|-------------|
> |3× extrapolation|5 GPU·hours|0.9465  | 62    | 65.00        |26.45      | 100     |
> |3× extrapolation with FastVideo|0.3 GPU· hours|0.9432|64  | 63.89|25.98 | 100|
> |4× extrapolation|8GPU·hours|0.9491|42|66.54|24.52|99.87|
> |4× extrapolation with FastVideo|0.5 GPU·hours|0.9399|40|62.24|24.83|96.32|

---

> ### Author Response · Authors · 2025-11-21
>
> ## Q3. The paper omits direct comparison to some alternative approaches for long video generation like FIFO-Diffusion.
>
> As discussed in **Common Concern 1**, our method aims to extend the effective training window of video diffusion transformers and is orthogonal to existing long-video approaches (e.g., FIFO, FreeNoise, sliding-window generation). On top of UltraViCo, these techniques can be further applied to additionally enhance long-term temporal consistency. We have incorporated the above discussion into the Experiments section of the revised paper.
>
>
> ## Q4. The theoretical analysis stops short of fully justifying the chosen solution. The idea of suppressing out-of-window attention is intuitively motivated and empirically validated, but there isn’t a formal framework proving it as an optimal or necessary fix.
>
> Thanks for the question. Based on softmax normalization, attenuating the logits of tokens sharpens the attention distribution and improves focus. This naturally raises two questions:
> (1) Which tokens should be attenuated?
> (2) How should the decay be applied?
>
> For (1), our ablations in Lines 297–299 show that attenuating only tokens outside the training window is the most effective strategy. This is consistent with the intuition that a token’s local neighbors contain the strongest correlations and should therefore remain unaltered.
>
> For (2), we experimented with multiple decay functions—including a simple constant decay and more complex linear or parabolic forms—and found only negligible differences (as shown in Fig. 7). For simplicity and clarity, we adopt a constant decay factor applied solely to tokens outside the training window.
>
> A more complete theoretical analysis of decay forms and their interaction with attention normalization is an interesting direction, and we will explore more principled decay formulations in future work.
>
>
> ## Q5. Does suppressing distant-frame attention risk any long-term coherence issues?
>
>
> As discussed in Q1 and **Common Concern 2**, over-aggressive decay can indeed cause long-term coherence issues. However, with appropriate choices of $\alpha$ and $\beta$, the distant-frame relations are attenuated but not removed, ensuring that essential scene information (e.g., background layout and character identity) remains preserved throughout the sequence. For example, on Wan under 4× extrapolation, motion dynamics increase by over 400% (11 → 47) while temporal consistency remains stable (0.9415 → 0.9484). This indicates that UltraViCo can enhance motion expressiveness without sacrificing long-term coherence.
>
>
> ## Q6. Have the author considered combining UltraViCo with other long-video generation techniques? Could one apply UltraViCo in a sliding-window generation to further improve consistency at the window boundaries, or use it alongside a method like FreeNoise to additionally boost temporal stability?
>
>
> Thank you for the insightful suggestion. As discussed in **Common Concern 1**, UltraViCo is indeed complementary to existing long-video generation techniques. Combining UltraViCo with approaches such as sliding-window generation or FreeNoise can further improve long-range temporal consistency. We have added an independent section in the revised paper to elaborate on this compatibility and potential integration. We have incorporated the above discussion into the Experiments section of the revised paper.
>
> ## Q7. Could the authors clarify why setting the first frame decay factor to a negative value to counteract blurring.
>
> Thank you for the careful review. We hypothesize that this is caused by the causal design of the video VAE, where the first frame is encoded independently and without temporal compression. As a result, it exhibits different statistical properties from subsequent frames and becomes more sensitive to perturbations. A similar phenomenon is also discussed in Self-Forcing [1]. We have included the above explanation in Appendix C.2 of the revised paper.
>
>
> [1] Self Forcing: Bridging the Train-Test Gap in Autoregressive Video Diffusion
>
>
> ## Q8. How should one choose the decay parameters for a new model or a different extrapolation ratio?
>
> As discussed in **Common Concern 2**, our detailed analysis of $\alpha$ and $\beta$ shows a consistent and robust pattern across models and extrapolation ratios:
> when $ \alpha $ ≥ 0.9 and $\beta$ ≥ 0.6, visual quality and motion dynamics improve substantially, while temporal consistency remains comparable to the baseline. Therefore, we recommend $\alpha$ = 0.9 and $\beta$ = 0.6 as a strong default, which already achieves improvements over the baseline. For users who prefer slightly stronger motion or stronger consistency, a light adjustment around the defalt is enough (e.g.  $\beta=0.8$  for higher consistency, $\alpha$ = 0.85 for stronger quality). We have incorporated the above discussion into the Experiments section of the revised paper.

---

### Official Review · Reviewer_apZp · 2025-10-31

**Soundness:** 3
**Presentation:** 3
**Contribution:** 3
**Rating:** 2
**Confidence:** 5

**Summary:**

This paper proposes a length-extension method for video generation models, aiming to generate longer videos without additional training. The authors achieve this by analyzing and modifying the attention mechanism. In the experiments, they compare their method with other training-free video extension approaches and demonstrate its effectiveness.

**Strengths:**

* This study follows a clear methodology: observation, analysis, improvement, and validation, with rigorous logic and well-structured presentation.
* Experiments demonstrate the effectiveness of the proposed method on Wan and HunyuanVideo, outperforming baseline approaches.

**Weaknesses:**

* The authors’ modifications to the computation process may introduce potential performance issues. Their improvements focus on the attention map, but in efficient attention implementations (e.g., FlashAttention-3), this matrix is not fully materialized. Consequently, the proposed changes could negatively impact computational efficiency.
* Generating long videos is no longer a major challenge, as several publicly available models already support segment-wise long-video generation—such as Wan2.2-S2V, Wan-Animate, and LongCat-Video. The authors should consider including these models in their baseline comparisons.

**Questions:**

Please refer to the weaknesses.

---

> ### Author Response · Authors · 2025-11-21
> **Author Response to Reviewer apZp**
>
> We thank reviewer apZp for the valuable comments. We address the concerns as follows.
>
> ## Q1. The proposed changes could negatively impact computational efficiency.
>
> We clarify that this is a misunderstanding. Our method **does not require materializing the full attention matrix** and can be seamlessly **integrated into efficient online attention kernels**. As stated in **Lines 375–377 of the original submission** and implemented in the `_attn_fwd_inner` function of **attn_qk_int8_per_block.py in the original submitted code**, UltraViCo has already been integrated into kernels such as SageAttention [1]. The complete procedure is summarized in **Algorithm 1** of the revised paper（Appendix D.1）. Moreover, as reported in **Rebuttal Table 4**, when built on top of FlashAttention [2] and SageAttention, UltraViCo incurs almost **no additional overhead in either latency or memory usage**. All measurements were obtained on A800 GPUs.
>
>
> **Rebuttal Table 4.** Runtime and memory comparison. Note that SageAttention is optimized for 4090-like architectures; on A800, its runtime is comparable to FlashAttention.
> | Model / Method              | Time (s / iter) | Memory (per GPU) |
> |-----------------------------|------------------|-------------------|
> | **HunyuanVideo (3× extrapolation)** |                  |                   |
> | Sageattention                 | 341.2            | 73188M            |
> | Sageattention + Ours          | 349.6            | 72346M            |
> | Flashattention                | 349.3            | 76030M            |
> | Flashattention  + Ours         | 355.3            | 75932M            |
> | **Wan (3× extrapolation)**                                       |
> | Sageattention                 | 32.13            | 24342M            |
> | Sageattention  + Ours         | 34.12            | 24342M            |
> | Flashattention               | 32.64            | 24349M            |
> | Flashattention + Ours        | 33.74            | 24346M            |
>
>
>
> [1] Sageattention: Accurate 8-bit attention for plug-and-play inference acceleration.
>
> [2] Flashattention: Fast and memory-efficient exact attention with io-awareness.
>
> ## Q2. Generating long videos is no longer a major challenge, as several publicly available models already support segment-wise long-video generation—such as Wan2.2-S2V, Wan-Animate, and LongCat-Video. The authors should consider including these models in their baseline comparisons.
>
>
> We clarify that our method is **orthogonal** to segment-wise long-video approaches and can **be integrated into** these pipelines to improve long-term temporal consistency. As shown in **Figure 9** in the revised paper, segment-wise methods such as Wan2.2-TI2V condition only on a few ending frames from the previous segment, ignoring content from early frames, which can lead to **character identity degradation** over long-term videos. By extending each segment’s effective context via UltraViCo, these methods can better **preserve character consistency across segments**.
>
> Furthermore, UltraViCo is designed as a **training-free** approach to any Video Diffusion Transformer across diverse tasks, whereas segment-wise long-video methods require **task-specific or model-specific training**. It is impractical to fine-tune a separate segment-wise long-video model for every community model or every controllable/editing scenario. As shown in Figure 1b in the original submitted paper, UltraViCo can be directly applied to controllable video synthesis and video editing without any retraining. We have incorporated the above discussion into the Experiments section of the revised paper.

---

> > ### Author Response · Authors · 2025-11-27
> > **Looking forward to further feedback**
> >
> > Dear Reviewer apZp,
> >
> > Thank you again for the great efforts and comments. We have carefully addressed the main concerns in detail. We believe the additional experiments, analysis, and explanation have significantly improved the quality and clarity of our submission.  We hope you might find the response satisfactory (similar as the other reviewers) and regard this as a sufficient reason to raise the score. As the discussion phase is about to close, we are very much looking forward to hearing from you about any further feedback. We will be very happy to clarify further concerns (if any).
> >
> > Best, Authors

---

### Official Review · Reviewer_ju43 · 2025-10-31

**Soundness:** 3
**Presentation:** 3
**Contribution:** 3
**Rating:** 8
**Confidence:** 3

**Summary:**

The paper tackles video length extrapolation in diffusion transformer–based text‑to‑video models, which suffer from repetition and quality loss beyond training length. It identifies attention dispersion as the root cause and introduces UltraViCo, a simple training‑free decay on out‑of‑window attention logits. This method extends extrapolation from 2x to 4x and significantly improves motion and visual quality across multiple models.

**Strengths:**

1) The paper is very well written; the figures are very helpful to understand the motivation behind the proposed method.
2) This work provides a compelling analysis that connects two failure modes (repetition and quality degradation) under the concept of attention dispersion, offering a deeper understanding of underpinnings of video diffusion transformer models.
3) The proposed solution UltraViCo is training-free, simple to implement, and integrates seamlessly with existing popular SOTA video DiTs.
4) Very good empirical results and comprehensive evaluation of the method and baselines.
5) The approach demonstrates applicability to downstream tasks like controllable video generation and video editing without retraining.

**Weaknesses:**

1) UltraViCo relies on manually chosen decay factors (alpha, beta) and harmonic band width (gamma), which differ across models. The paper does not provide a fully automated procedure for selecting these values, raising concerns about generalization to unseen architectures.

**Questions:**

1) How can alpha, beta, gamma be selected without manual tuning? Could you propose a lightweight heuristic or adaptive scheme that works across unseen models?
2) Why stop at 4x only? How does model perform beyond this threshold? If not, why do you think the method is not suitable for extreme long video generation task?
3) What do you think about other works/directions that are trying to tackle long video generation (e.g., Diffusion Forcing, Frame Pack)? Do you think augmenting/optimizing attention maps (the approach taken in this paper) is a more promising direction as opposed to others?

---

> ### Author Response · Authors · 2025-11-21
> **Author Response to Reviewer ju43**
>
> We sincerely thank Reviewer ju43 for the recognition of our work. The further questions are addressed as follows.
>
> ## Q1. UltraViCo relies on manually chosen decay factors (alpha, beta) and harmonic band width (gamma), which differ across models. How can alpha, beta, gamma be selected without manual tuning? Could you propose a lightweight heuristic or adaptive scheme that works across unseen models?
>
> As discussed in **Common Concern 2**, our detailed analysis of $\alpha$ and $\beta$ shows a consistent and robust pattern across models and extrapolation ratios: when $\alpha$ ≥ 0.9 and $\beta$ ≥ 0.6, visual quality and motion dynamics improve substantially, while temporal consistency remains comparable to the baseline. Therefore, we recommend $\alpha$ = 0.9 and $\beta$ = 0.6 as default settings, which already outperform the baseline. For $\gamma$ , we set it such that the width of $P_{\text{risk}}$ is approximately one-third of the dominant repetition period (i.e., $2\gamma+1\approx \frac{1}{3} L$), which reliably covers the region where periodic repetition may occur. We have incorporated the above discussion into the Experiments section of the revised paper.
>
> ## Q2. Why stop at 4x only? How does model perform beyond this threshold? If not, why do you think the method is not suitable for extreme long video generation task?
>
> Thank you for the insightful comments. A larger extrapolation ratio introduces substantially more tokens, which makes the attention distribution increasingly flat. Due to the normalization effect of softmax, re-concentrating attention back onto the original training window requires an excessively strong decay factor, and such over-correction can harm temporal consistency.
>
> Notably, our method primarily extends the effective context window of any video diffusion transformer, and is therefore complementary to existing long-video approaches (e.g., FIFO, FreeNoise, Sliding Window in **Common Concern 1**). UltraViCo can be combined with these techniques to achieve extremely long video generation and further improve long-term temporal consistency. We have incorporated the above discussion into the Experiments section of the revised paper.
>
> ## Q3. What do you think about other works/directions that are trying to tackle long video generation (e.g., Diffusion Forcing, Frame Pack)? Do you think augmenting/optimizing attention maps (the approach taken in this paper) is a more promising direction as opposed to others?
>
>
> Thank you for the comments. We view recent efforts such as Diffusion Forcing, FiFO and Frame Pack as valuable and complementary directions for long video generation. Our work focuses on a different but orthogonal goal: extending the effective training window of any video diffusion transformer in a training-free manner, without modifying model parameters. This allows UltraViCo to be directly applied to downstream tasks without retraining (see Figure 1b), and to be combined with existing long-video techniques to further enhance temporal consistency and scale to even longer videos (see **Common Conern 1**). We have incorporated the above discussion into the Experiments section of the revised paper.

---

### Author Response · Authors · 2025-11-21
**Common Concerns from reviewers**

We address the common concerns here and post a point-to-point response to each reviewer as well.

## Q1. Connection with existing long-video generation methods (from reviewer ju43, Tf16 and 3Ey3).

We clarify that UltraViCo aims to extend the effective training window of video diffusion transformers and is therefore **orthogonal** to existing long-video generation techniques such as FreeNoise, FIFO, and sliding-window. As demonstrated in **Rebuttal Table 1**, **enlarging the context window via UltraViCo consistently improves the long-term temporal consistency of these methods**, without negatively affecting other performance metrics. In Rebuttal Table 1, all methods follow the same evaluation setup (6× extrapolation for 30-second videos on Wan), where UltraViCo extends the base model’s training window by 3×. We have incorporated the above discussion into the Experiments section of the revised paper.



**Rebuttal Table 1**:  Application of UltraViCo on existing long-video methods.
| Method              | Consistency $\uparrow$ | Dynamics $\uparrow$  | Quality $\uparrow$  | Overall $\uparrow$  |
|---------------------|----------------|-------------|-------------|-------------|
| Sliding Window                | 0.8478         | 56          | 62.94       | 23.57      |
| Sliding Window  + UltraViCo       | **0.9183**         | 54          | 62.85       | 23.95      |
| FreeNoise           | 0.9243         | 38          | 63.09       | 23.75      |
| FreeNoise + UltraViCo  | **0.9431**         | 41          | 62.12       | 23.92      |
| FIFO                | 0.9131         | 53          | 61.31       | 23.81      |
| FIFO + UltraViCo      | **0.9319**         | 51          | 63.09       | 24.24      |





## Q2. Choice of $\alpha$ and $\beta$ (from reviewer ju43, Tf16 and 3Ey3).

Building on the detailed sensitivity study in **Rebuttal Table 2,3 and Figure 8** in the revised paper, we observe a clear and stable pattern: **when $\alpha$ ≥ 0.9 and $\beta$ ≥ 0.6, visual quality and motion dynamics improve substantially, while temporal consistency remains comparable to the baseline**. Therefore, $\alpha$ = 0.9 and $\beta$ = 0.6 serve as robust default choices, which already achieves improvements over the baseline. For users who prefer slightly stronger motion or stronger consistency, a light adjustment around the defalt is enough (e.g. $\beta$ =0.8 for higher consistency, $\alpha$ = 0.85 for stronger quality). Notably, dynamics and consistency naturally trade off; although $\alpha$ ≥ 0.9 and $\beta$ ≥ 0.6 may has slight decrease consistency, we find consistency above 0.94 remains visually well (e.g., Wan’s training-horizon consistency is ≈0.95). The detailed sensitivity analysis is as follows. We have incorporated the above discussion into the Experiments section of the revised paper.


**Rebuttal Table 2**: Sensitivity analysis of $\alpha$ on Hunyuan at 3x extraplation. We set $\beta$ the same as $\alpha$. In other words, a single decay factor is shared globally.
| $\alpha$   | Consistency $\uparrow$    | Dynamics $\uparrow$ | Quality $\uparrow$     | Overall $\uparrow$   | NoRepeat $\uparrow$  |
|------------|--------|-------|--------|--------|--------|
| 1     | 0.9795     | 16      | 51.85   | 21.62   | 53.17              |
| 0.95 | 0.9663 | 25  | 54.92  | 24.07 | 100    |
| 0.9  | 0.9647 | 32  | 57.53  | 26.25 | 93.34  |
| 0.85 | 0.9298 | 68  | 69.93  | 26.89 | 99.53  |
| 0.8   | 0.9231 | 73  | 70.35  | 26.96 | 100    |



**Rebuttal Table 3**: Sensitivity analysis of $\beta$ on Hunyuan at 3x extraplation. We set $\alpha=0.9$ across all settings.
| $\beta$       | Consistency $\uparrow$ | Dynamics $\uparrow$ | Quality $\uparrow$ | Overall $\uparrow$ | NoRepeat $\uparrow$ |
|---------|------------|---------|---------|---------|----------------|
| 1     | 0.9716     | 28      | 55.23   | 24.52   | 57.42              |
| 0.9     | 0.9647     | 32    | 57.53   | 26.25  | 93.34          |
| 0.8     | 0.9510     | 45    | 59.35   | 26.42  | 97.25          |
| 0.75    | 0.9496     | 51    | 62.11   | 26.98  | 95.77          |
| 0.6     | 0.9465     | 62      | 65.00      | 26.45  | 100              |
| 0.45    | 0.9349     | 65    | 68.34   | 26.99  | 100            |
| 0.3     | 0.9318     | 66    | 70.45   | 26.98  | 100            |






Specifically:

(1) α sensitivity:
  - When α ≥ 0.9, both quality and motion dynamics improve significantly, while temporal consistency remains comparable.
  - When α < 0.9, consistency drops sharply.

(2) β sensitivity:
  - When β ≥ 0.6, quality and dynamics remain high, and consistency stays comparable.
  - When β < 0.6, consistency degrades significantly.

These trends justify our recommended default of $\alpha$ = 0.9 and $\beta$ = 0.6. We thank the reviewer for the insightful comments, which have helped improve the clarity and quality of our work. We have added the above discussion in Section 4 of the revised paper.

---

### Meta-Review · Area_Chair_ASDu · 2026-01-07

**Summary:**

The paper proposes UltraViCo, a training-free approach to improve the extrapolation capabilities of video diffusion models. It begins by analyzing the failure modes of these models and identifies attention dispersion as the primary cause. To address this, the paper introduces a straightforward method to enhance extrapolation through attention concentration. The effectiveness of the proposed approach is validated on multiple publicly available video diffusion models.

The initial reviews are mixed. Major concerns are about its connection with other approaches, such as Sliding Window, FIFO, and segment-wise long-video models. In the rebuttal, the authors have addressed these concerns through comprehensive experiments. Given the technical contribution and its high-quality results, I would recommend acceptance of this work. I encourage the authors to incorporate the rebuttal tables in their next version.

**Reviewer Concerns:**

Most of the major concerns are addressed in the rebuttal.

**Reviewer Scores:**

8, 2, 6, 4 -> 8, 2, 6, 6

---

### Decision · Program_Chairs · 2026-01-26

Accept (Poster)